# Genetic loci for lung function in Japanese adults with adjustment for exhaled nitric oxide levels as airway inflammation indicator

Mitsuhiro Yamada [1,11], Ikuko N. Motoike[2,11], Kaname Kojima[2,11], Nobuo Fuse [2], Atsushi Hozawa[3], Shinichi Kuriyama [3], Fumiki Katsuoka [2], Shu Tadaka [2], Matsuyuki Shirota[2], Miyuki Sakurai[2], Tomohiro Nakamura[4], Yohei Hamanaka[2], Kichiya Suzuki[5], Junichi Sugawara[6], Soichi Ogishima [4,7], Akira Uruno[2], Eiichi N. Kodama[6], Naoya Fujino [1], Tadahisa Numakura [1], Tomohiro Ichikawa[1], Ayumi Mitsune[1], Takashi Ohe[1], Kengo Kinoshita [2,7,8], Masakazu Ichinose[9], Hisatoshi Sugiura [1] & Masayuki Yamamoto [2,10 ✉]

Lung function reflects the ability of the respiratory system and is utilized for the assessment of respiratory diseases. Because type 2 airway inflammation influences lung function, genome wide association studies (GWAS) for lung function would be improved by adjustment with an indicator of the inflammation. Here, we performed a GWAS for lung function with adjustment for exhaled nitric oxide (FeNO) levels in two independent Japanese populations. Our GWAS with genotype imputations revealed that the *RNF5/AGER* locus including *AGER* rs2070600 SNP, which introduces a G82S substitution of AGER, was the most significantly associated with $FEV_1/FVC$. Three other rare missense variants of AGER were further identified. We also found genetic loci with three candidate genes (*NOS2, SPSB2* and *RIPOR2*) associated with FeNO levels. Analyses with the BioBank-Japan GWAS resource revealed genetic links of FeNO and asthma-related traits, and existence of common genetic background for allergic diseases and their biomarkers. Our study identified the genetic locus most strongly associated with airway obstruction in the Japanese population and three genetic loci associated with FeNO, an indicator of type 2 airway inflammation in adults.

[1] Department of Respiratory Medicine, Tohoku University Graduate School of Medicine, Sendai, Japan. [2] Department of Integrative Genomics, Tohoku Medical Megabank Organization, Tohoku University, Sendai, Japan. [3] Department of Preventive Medicine and Epidemiology, Tohoku Medical Megabank Organization, Tohoku University, Sendai, Japan. [4] Department of Health Record Informatics, Tohoku Medical Megabank Organization, Tohoku University, Sendai, Japan. [5] Department of Biobank, Tohoku Medical Megabank Organization, Tohoku University, Sendai, Japan. [6] Department of Community Medical Supports, Tohoku Medical Megabank Organization, Tohoku University, Sendai, Japan. [7] Advanced Research Center for Innovations in Next-Generation Medicine, Tohoku University, Sendai, Japan. [8] Department of System Bioinformatics, Tohoku University Graduate School of Information Sciences, Sendai, Japan. [9] Osaki Citizen Hospital Academic Center, Osaki, Japan. [10] Department of Medical Biochemistry, Tohoku University Graduate School of Medicine, Sendai, Japan. [11]These authors contributed equally: Mitsuhiro Yamada, Ikuko N. Motoike, Kaname Kojima. ✉email: masiyamamoto@med.tohoku.ac.jp

Measures of lung function such as forced expiratory volume in one second ($FEV_1$) and the ratio of $FEV_1$ to forced vital capacity ($FEV_1$/FVC) reflect the condition of the respiratory system. Lung function is one of the predictors for mortality in the general population[1–3], and is also used for both the diagnosis and assessment of respiratory diseases including chronic obstructive pulmonary disease (COPD). COPD is an important cause of death all over the world[4,5]. The development of COPD is mainly influenced by environmental factors, and cigarette smoking is the most common cause of COPD[6,7]. However, it has been suggested that the pathogenesis of COPD could also be significantly influenced by genetic factors, because some people develop COPD while others do not, even though both have similar smoking histories[8–11]. We, therefore, believe that COPD is one of the best target diseases for developing a personalized healthcare algorithm based on genome information.

Over the past decade, world-wide multi-centre genome-wide association studies (GWASs) have been performed, and these studies have identified genetic loci that show significant associations with the lung function, COPD affection and COPD-related phenotypes such as emphysema[12–20]. However, no large-scaled GWAS study for identifying genetic variants associated with lung function in an adult population has been conducted in Japan. Recent whole-genome sequence data in the Northeast Asian Reference Database (NARD) revealed that the ancestral composition of Japanese populations is different from those of other East Asian populations including Korean and Han Chinese[21]. This information leads us to the notion that it is worth performing GWAS of lung function for Japanese individuals, as Japanese people are relatively different from the other East Asians in terms of ancestral compositions. Besides, the sizes of lung-function GWAS in East Asia appeared to be rather limited, at best <8900, we wish to perform much larger-scale GWAS than those reported previously[22,23]. Further, the clinical characteristics of COPD patients in East Asia including Japan seems to be different from those of the patients in the US and Europe, where most of the previous GWASs were performed[24–29]. For instance, proportions of the patients having emphysema-dominant phenotype are greater in Japanese[27,29] and Korean[25] COPD populations than those in the US and Europe COPD populations. Frequencies of exacerbation in Japanese and Korean COPD populations are smaller than those in the US and Europe COPD populations[24,26,28]. Based on these reasons, we decided to explore genetic variants associated with the lung function measures in the Japanese population, as it could provide meaningful information for elucidating the genetic vulnerability for pulmonary diseases including COPD in the world.

The Tohoku Medical Megabank (TMM) has been established in the Miyagi and Iwate Prefectures in Japan. In the Miyagi Prefecture, the Tohoku Medical Megabank Organization (ToMMo) is operating at Tohoku University, while in the Iwate Prefecture, the Iwate TMM Organization is operating at Iwate Medical University. Two genome cohort studies have been conducted by TMM[30–33], which were strategically designed to assess both the long-term impact of the Great East Japan Earthquake on disaster victims and the gene-environmental interactions on the incidence of major common diseases[31]. Of the two cohort studies, one is a community-based adult cohort study, named the TMM Community-Based Cohort Study (CommCohort Study), while the other is a birth and three-generation cohort study, named the TMM Birth and Three-Generation Cohort Study (BirThree Cohort Study). The latter also aims to assess disease risk throughout a person's lifespan in multiple generations by means of a long-term follow-up[30]. The TMM Project aims to establish an integrated biobank, sharing both biological samples and genome-omics information along with routine health examination data[33,34]. To this end, we have been conducting a large-scale ethnic-specific SNP array analysis[32], through which we are attempting to elucidate the genetic vulnerabilities of common diseases including COPD to enable personalized health care[32].

In this study, therefore, we performed GWAS for lung function ($FEV_1$ and $FEV_1$/FVC) in the two ToMMo independent cohorts from the Miyagi prefecture. We utilized the ToMMo CommCohort Study for the Discovery Stage and ToMMo BirThree Cohort Study for the Validation Stage to elucidate the genetic vulnerabilities for respiratory diseases including COPD. In this study, we wished to perform GWAS for $FEV_1$/FVC, an indicator for airway obstruction, with or without adjustment by the levels of exhaled nitric oxide (FeNO), an indicator of type 2 airway inflammation[35]. Because type 2 airway inflammation, which involves accumulation of eosinophils, mast cells, basophils, Th2 cells, type 2 innate lymphoid cells (ILC2s) and IgE-producing B cells, and type 2 cytokines (IL-4, IL-5, and IL-13), could also influence lung function, especially airway obstruction, we tried to find the genetic factors related to $FEV_1$/FVC regardless of the existence of airway type 2 inflammation. To explore genetic factors related to airway type 2 inflammation, we also performed a large-scaled GWAS for FeNO. To our best knowledge, our GWAS for FeNO is the largest one for adults in terms of the number of subjects.

## Results

**Characteristics of study population**. We analysed 14,061 genotyped participants measured for lung function and FeNO in the ToMMo CommCohort Study for the Discovery Stage analysis after quality control. We excluded 47 subjects from the study because of inconsistent sex declaration by themselves with those by genotyping, and seven subjects whose identical twins also participated in the study. We also examined 5661 genotyped participants in the ToMMo BirThree Cohort for the Validation Stage analysis after quality control from which 11 subjects were excluded for similar reasons.

The main demographical characteristics of these two study cohorts are summarized in Supplementary Table 1. About 30% of the participants were male in both cohorts. The median age was 67.0 years in the ToMMo CommCohort, whereas it was 39.1 years in the ToMMo BirThree Cohort. 32.7% and 40.9% of the participants of the ToMMo CommCohort and ToMMo BirThree Cohort were ever-smokers, respectively.

**GWAS for association with the measures of lung function at the Discovery Stage**. The study design for investigating genetic factors that affect lung function is shown in Fig. 1. To explore the genetic locus associated with lung function measures ($FEV_1$/FVC and $FEV_1$), we first undertook a genome-wide analysis for 8,587,571 genotyped or imputed genetic variants in the participants of the ToMMo CommCohort Study as the Discovery Stage. We adjusted the $FEV_1$/FVC measures for age, sex, smoking status as covariates and the $FEV_1$ measures for age, sex, height, and smoking status as covariates. We also performed GWAS for $FEV_1$/FVC with or without the adjustment by FeNO, because we wished to examine the effect of the adjustment in terms of exploring the genetic factors for airway obstruction regardless of the existence of airway type 2 inflammation (Supplementary Table 2).

The quantile-quantile plots of the GWAS at the Discovery Stage suggested the presence of multiple variants associated with $FEV_1$/FVC, whereas fewer variants with weaker effects were suggested for $FEV_1$ (Fig. 2a, b). We observed independent regions of significant association at four loci (144 variants; 122 SNPs plus 22 insertion/deletions with $P < 5 \times 10^{-8}$) for $FEV_1$/FVC (Fig. 2c, Table 1, Supplementary Data 1) and two loci (58 variants; 51 SNPs plus 7 insertion/deletions) for $FEV_1$. The four loci for $FEV_1$/FVC were 6p21.32 in *RNF5/AGER/PPT2-EGFL8*, 2q37.3 in *LINC01940~HDAC4*, 4q22.1 in *FAM13A* and 1p21.2 in *PLPPR4~LINC01708*. The

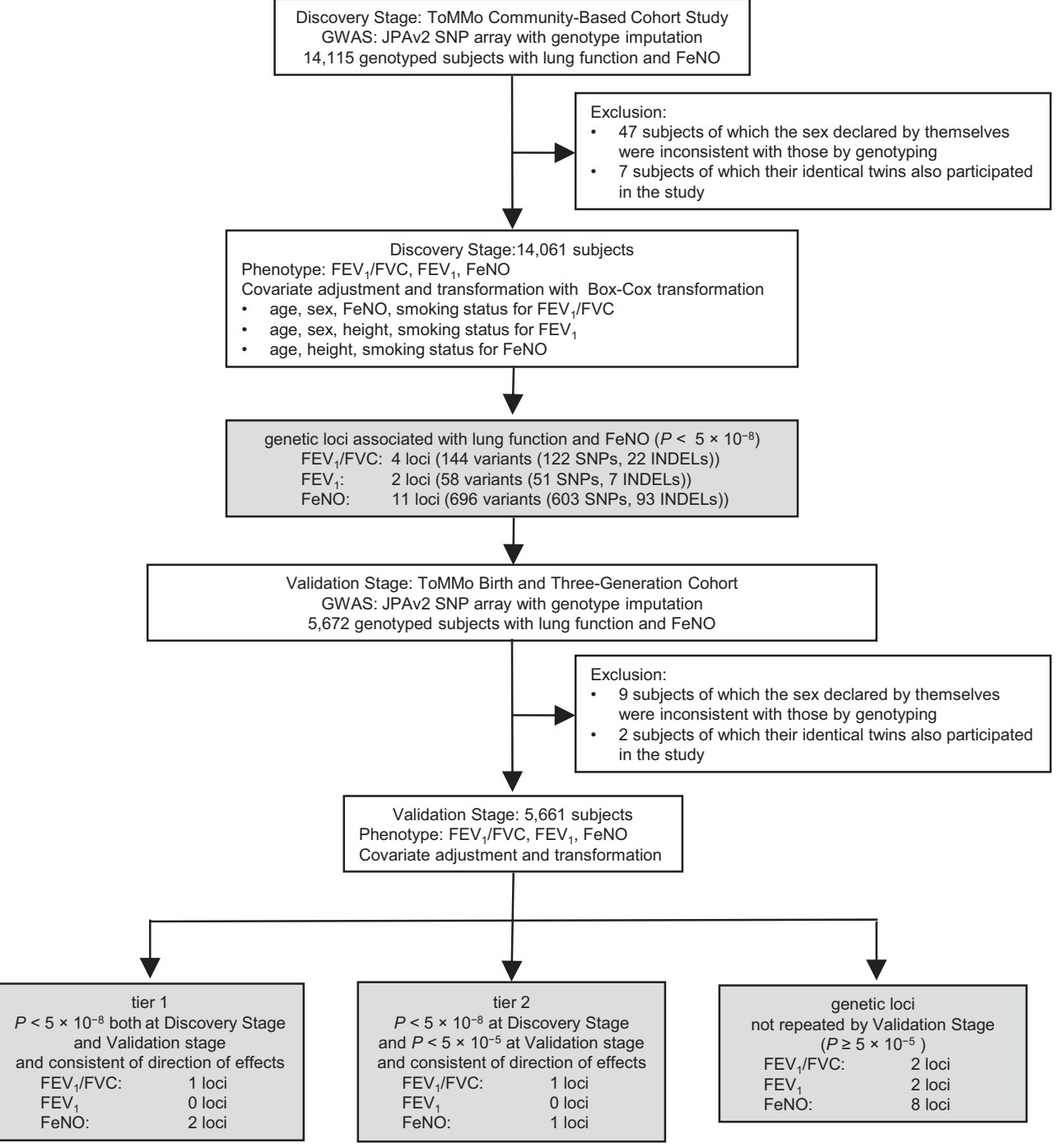

**Fig. 1 Study design for investigating genetic factors that determine lung function and FeNO.** The study design for investigating the genetic factors that determine lung function and FeNO is shown. Note that tier 1 loci have $P < 5 \times 10^{-8}$ both at the Discovery Stage and Validation Stage, while tier 2 signals have $P < 5 \times 10^{-8}$ at the Discovery Stage and $P < 5 \times 10^{-5}$ at the Validation Stage with consistent directions of effect. FeNO exhaled nitric oxide, $FEV_1$ forced expiratory volume in one second, FVC forced vital capacity, SNP single nucleotide polymorphism, INDEL insertion or deletion, JPAv2 Japonica array version 2.

two loci for $FEV_1$ were 4q31.21 in *GYPA~HHIP-AS1* and 5p12 in *FGF10* (Fig. 2d, Table 1, Supplementary Data 2).

Contrary to our expectation, the result of GWAS with FeNO adjustment is similar to that without the adjustment, although the *p*-value of genetic locus 4q22.1 in *FAM13A* is lower in the GWAS with FeNO adjustment than that without adjustment (Supplementary Fig 1, Supplementary Table 3), indicating that the adjustment with FeNO did not influence substantially for the GWAS results of $FEV_1$/FVC. We also performed the GWAS of

$FEV_1$ with the adjustment by FeNO. The result was not significantly different from that without the adjustment with FeNO (Supplementary Table 4).

**Association of the *RNF5/AGER* locus is most significant with $FEV_1$/FVC.** To validate potential associations between lung function and the genetic locus suggested at the Discovery Stage, we then conducted a genome-wide analysis for 8,595,665

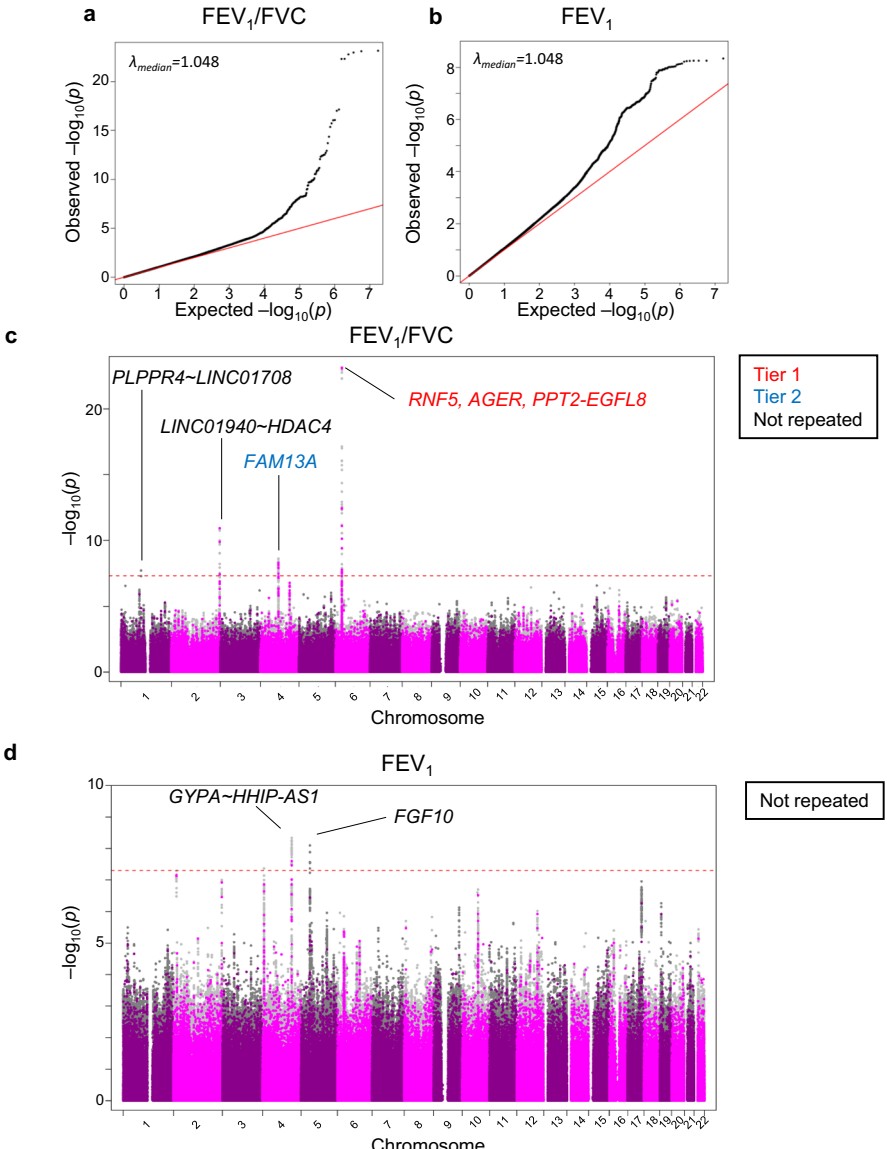

**Fig. 2 GWAS for association with the measures of lung function at the Discovery Stage.** GWAS was conducted for the association with the measures of lung function at the Discovery Stage. Quantile-quantile (QQ) plots of the observed versus expected log10($P$) values in $FEV_1/FVC$ ratio (**a**) and $FEV_1$ (**b**) are shown with the inflation factor ($\lambda$). The Manhattan plots of GWAS for the association with the $FEV_1/FVC$ ratio (**c**) and $FEV_1$ (**d**) show the chromosomal positions of variants exceeding the genome-wide significance threshold ($P < 5 \times 10^{-8}$ as indicated by the dotted red line). Gene names in red correspond to tier 1 loci ($P < 5 \times 10^{-8}$ both in the Discovery Stage and Validation Stage); gene names in blue correspond to tier 2 loci ($P < 5 \times 10^{-8}$ in the Discovery Stage and $P < 5 \times 10^{-5}$ in the Validation Stage with consistent directions of effect); gene names in black correspond to genetic loci that had $P < 5 \times 10^{-8}$ in Discovery Stage but not repeated in the Validation Stage. Coloured circles mean the variants that were directly analysed. Grey circles mean the variants that were detected by genotype imputation. GWAS genome-wide association study, $FEV_1$ forced expiratory volume in one second, FVC forced vital capacity.

genotyped or imputed genetic variants in the participants of ToMMo BirThree Cohort as the Validation Stage (Fig. 1, Supplementary Fig. 2a, b). Based on these two stage analyses, we set up three tiers. Tier 1 loci had $p < 5 \times 10^{-8}$ both at the Discovery Stage and Validation Stage. Tier 2 loci had $p < 5 \times 10^{-8}$ at the Discovery Stage and $p < 5 \times 10^{-5}$ at the Validation Stage with consistent directions of effect. Tier 3 loci had $p < 5 \times 10^{-8}$ at the Discovery Stage but was not repeated in the Validation Stage (Fig. 1).

Out of four loci identified at the Discovery Stage for $FEV_1/$ FVC, 6p21.32 in *RNF5/AGER/PPT2-EGFL8* also had an association $p < 5 \times 10^{-8}$ at the Validation Stage with consistent directions of effect (tier 1; Table 1, Supplementary Fig 2c,

Supplementary Data 3). 4q22.1 in *FAM13A* had an association $p < 5 \times 10^{-5}$ with consistent directions of effect at the Validation Stage (tier 2; Table 1, Supplementary Fig 2c, Supplementary Data 3). For $FEV_1$, in none of the loci was the association repeated at the Validation Stage (Table 1, Supplementary Fig. 2d). Closer examinations for the *AGER* and *FAM13A* genes were conducted (Fig. 3, 4).

The strongest association with $FEV_1/FVC$ at both stages was at 6p21.32 with two top variants at the Discovery Stage, an exonic nonsynonymous SNP rs2070600 in *AGER* and intronic insertion/ deletion rs41315238 in *RNF5* (Table 1, Supplementary Data 1, Supplementary Data 3, Supplementary Table 5). *RNF5* rs41315238 is in near complete linkage disequilibrium with

**Table 1 Loci associated with lung function.**

| Pheno-type | Locus | Gene | Sentinel SNP | Function | Ref | Alt | EA | Imputed | Discovery stage | | | | Validation stage | | | | Verified | Reported |
|---|---|---|---|---|---|---|---|---|---|---|---|---|---|---|---|---|---|---|
| | | | | | | | | | EA freq. | Beta | SE | P | EA freq. | Beta | SE | P | | |
| FEV$_1$/FVC | 6p21.32 | RNF5 | rs4315238 | intronic | AAG | A | A | N | 0.1513 | 0.1542 | 0.0155 | 7.2 E-24 | 0.1510 | 0.1265 | 0.0234 | 7.9 E-08 | Tier 2 | N |
| | | AGER | rs2070600 | exonic | C | T | T | N | 0.1493 | 0.1548 | 0.0156 | 8.1 E-24 | 0.1493 | 0.1293 | 0.0235 | 4.8 E-08 | Tier 1 | Y |
| | | PPT2-EGFL8 | rs10947233 | ncRNA_intronic | G | T | T | Y | 0.1509 | 0.1516 | 0.0155 | 4.9 E-24 | 0.1507 | 0.1289 | 0.0234 | 4.6 E-08 | Tier 1 | Y |
| | 4q22.1 | FAM13A | 4:89834173_GGAAGAA_GGAA | intronic | GGA-AGAA | GGAA | GGAA | Y | 0.3997 | 0.0666 | 0.0114 | 2.5 E-23 | 0.3982 | 0.0838 | 0.0175 | 1.5 E-06 | Tier 2 | Y |
| | 2q37.3 | LINC01940-HDAC4 | rs36119524 | Inter-genic | C | T | T | Y | 0.2767 | 0.0841 | 0.0124 | 1.2 E-11 | 0.2846 | 0.0614 | 0.0187 | 1.1 E-03 | N | Y |
| | 1p21.2 | PLPPR4-LINC01708 | 1:99809254_TA_T | Inter-genic | TA | T | T | Y | 0.0349 | 0.1783 | 0.0314 | 2.0 E-08 | 0.0355 | -0.0245 | 0.0469 | 5.4 E-01 | N | Y |
| FEV$_1$ | 4q31.21 | GYPA-HHIP-AS1 | rs13107665 | Inter-genic | A | G | G | Y | 0.6770 | 0.0461 | 0.0079 | 4.6 E-09 | 0.6841 | 0.0415 | 0.0107 | 1.0 E-04 | N | N |
| | 5p12 | FGF10 | 5:44381582_A_AC | intronic | A | AC | AC | Y | 0.1163 | 0.0671 | 0.0016 | 8.0 E-09 | 0.1191 | 0.0319 | 0.0155 | 4.2 E-02 | N | Y |

Covariate adjustment with age, sex, FeNO, and smoking status for FEV$_1$/FVC; with age, sex, height, and smoking status for FEV$_1$.
Ref reference allele, Alt alternative allele, EA effect allele, EA freq. EA frequency, SE standard error, ncRNA non-coding RNA.

*AGER* rs2070600 (Supplementary Fig. 3). The regional association plot at 6p21.32 (Fig. 3a) also suggested that the probable candidate genes at this locus for the association with FEV$_1$/FVC measure is *AGER*, which codes an advanced glycosylation end product-specific receptor (AGER), a member of the immunoglobulin superfamily of cell surface receptors[36]. The protective minor T allele of SNP rs2070600 was associated with higher FEV$_1$/FVC ($p = 8.1 \times 10^{-24}$), with the highest mean FEV$_1$/FVC among TT homozygotes at the Discovery Stage (Fig. 4a).

Another salient association with FEV$_1$/FVC at both stages was 4q22.1 in *FAM13A* (Table 1, Fig. 3b). *FAM13A* encodes an intracellular adapter protein, which interacts with protein phosphatase 2A/β-catenin complex and promotes the degradation of β-catenin[37]. The protective minor GGAA allele of variant 4:89834173_GGAAGAA_GGAA was associated with higher FEV1/FVC ($p = 2.5 \times 10^{-9}$), with the highest mean FEV1/FVC among GGAA homozygotes (Fig. 4b).

We also performed a series of GWAS for lung function with the adjustment by the principal components (PC1~10). The results including significantly associated loci were not significantly different between the GWAS with and without the adjustment by the principal components (Supplementary Figs. 4, 5, 6, and 7).

To explore the function of these identified two loci for FEV$_1$/FVC, we utilized the Genotype-Tissue Expression (GTEx) Analysis V8 Fine-Mapping cis-eQTL data of lung tissue and whole blood (https://gtexportal.org/home/datasets) to find out the genes that expressions are related to the haplotypes of identified genetic variants in the two loci. The analyses revealed that the genetic variants at 6p21.32 is related to the expression of the genes including *AGER*, whereas the genetic variants at 4q22.1 is related to the expression of *FAM13A* (Supplementary Fig. 8). These results further reinforce our findings of GWAS, as the identified genetic variants affect the traits through the expression of candidate genes such as *AGER* and *FAM13A*.

**A common to rare approach identified a salient missense variant of AGER**. To explore the functional significance of exonic nonsynonymous SNP rs2070600 in *AGER*, we examined the variant position in the 3D structure of the AGER complex with S100A6 protein (Protein Data Bank ID: 4YBH[38]). Two extracellular domains of AGER bind to a homodimer of S100A6 protein, a protein with two EF-hand calcium-binding motifs, forming a tetrameric complex (Fig. 5a, b). The SNP rs2070600 in AGER introduces a G82S substitution of AGER in one of its immunoglobulin domains. As the Cα atom of G82 is in close contact with the side chain of L49 within 3.7 Å, an enlarged side chain of serine introduced to this position can affect the molecular function of the receptor, which may be responsible for the fact that a G82S substitution of AGER is associated with higher FEV$_1$/FVC (Fig. 5c).

Because our GWAS suggested that the exonic SNP of AGER significantly affected the lung function, we were interested in whether the general population harbours other non-synonymous genetic variants, which may influence the molecular function of AGER. Therefore, we next investigated our genomic allele reference panel, 4.7KJPN[39] and identified three other rare missense variants of AGER: G95R, R114Q and R244H. The first two are located in the same immunoglobulin domain in which G82 exists. In contrast, the R244 residue is located in the interface to S100A6. The R244 residue of AGER forms a cation-π interaction with F70 of S100A6 and binds to a chloride ion (Fig. 5d), demonstrating that R244H substitution impairs the interaction between an AGER and a S100A6, because two AGER subunits interact indirectly with each other via the S100A6 dimer. Thus, this R244H substitution seems to inhibit the dimerization and signalling of the AGER (Fig. 5e), which affects the biological

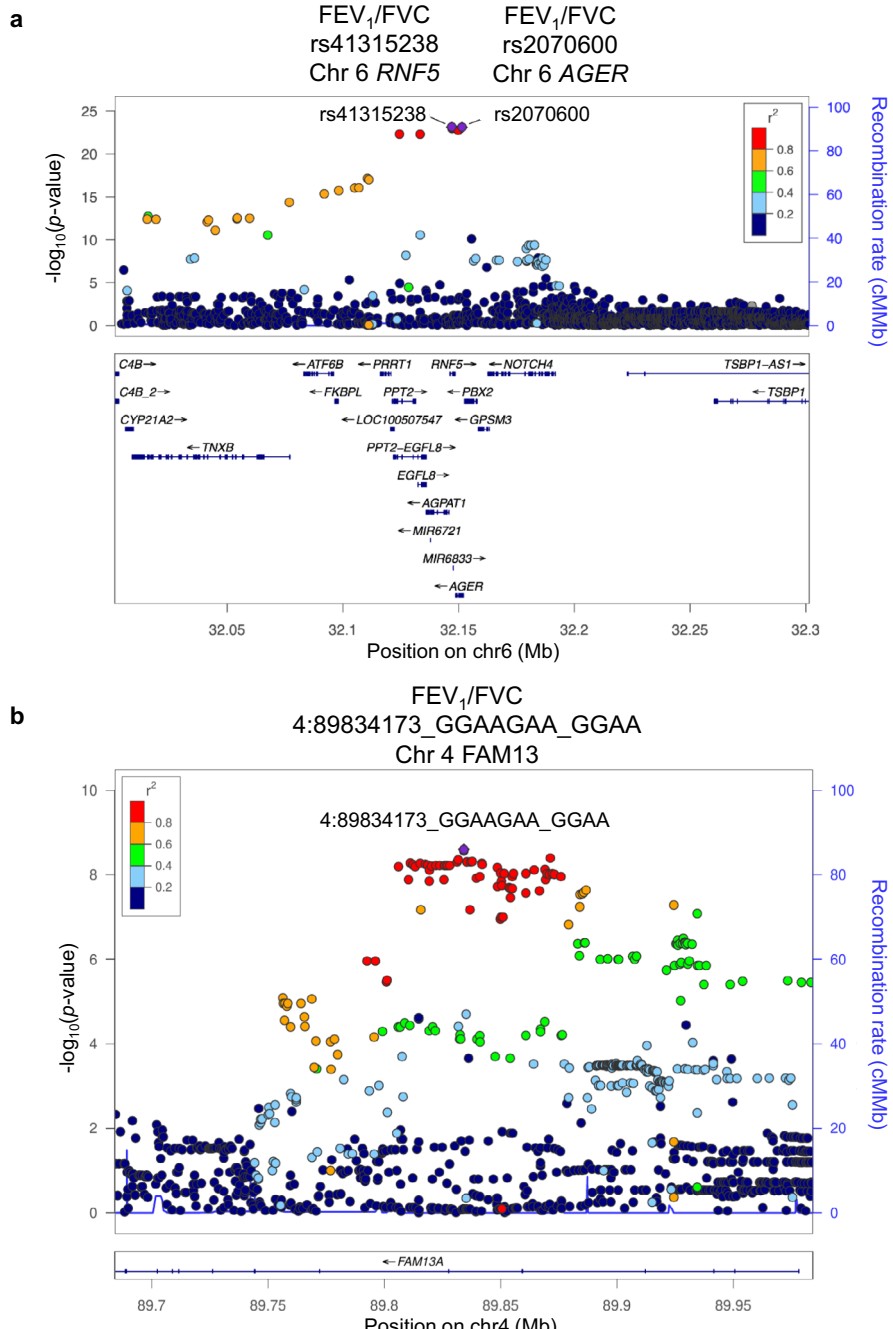

**Fig. 3 Regional association plots of the loci associated with FEV₁/FVC at both Discovery and Validation Stages. a**, **b** Statistical significance of each genetic variant on the −log10 scale as a function of chromosome position (hg19) at the Discovery Stage alone. The sentinel variant at each locus is shown in purple. The correlations ($r^2$) of each of the surrounding variants to the sentinel variant are shown in the indicated colours. In Fig. 4a, the correlations to rs2070600 are shown. The locus shown in Fig. 4a is a tier 1 locus that showed an association with FEV₁/FVC with $P < 5 \times 10^{-8}$ both at the Discovery Stage and Validation Stage. The locus in Fig. 4b is tier 2 locus that showed an association with FEV₁/FVC with $P < 5 \times 10^{-8}$ at the Discovery Stage and $P < 5 \times 10^{-5}$ at the Validation Stage with consistent directions of effect. FEV₁ forced expiratory volume in one second, FVC forced vital capacity.

functions of the AGER complex. We propose that the spectrum of AGER variants, from common to rare alleles, contributes to the lung function in the general population.

**GWAS for association with the levels of FeNO at Discovery Stage.** To explore the genetic locus associated with the levels of FeNO, we also performed a large-scale GWAS exploiting FeNO measures, as FeNO has been proposed as a noninvasive marker of

type 2 airway inflammation[35]. We first conducted a genome-wide analysis in the participants of the ToMMo CommCohort Study as the Discovery Stage (Fig. 1). We adjusted the FeNO measures for age, height, smoking status as covariates because these factors were suggested to be related to the levels of FeNO[40,41].

It should be noted that the quantile-quantile plots of the GWAS at the Discovery Stage suggested the presence of multiple significant variants associated with FeNO (Fig. 6a). Indeed, we observed ten independent regions of significant association (696

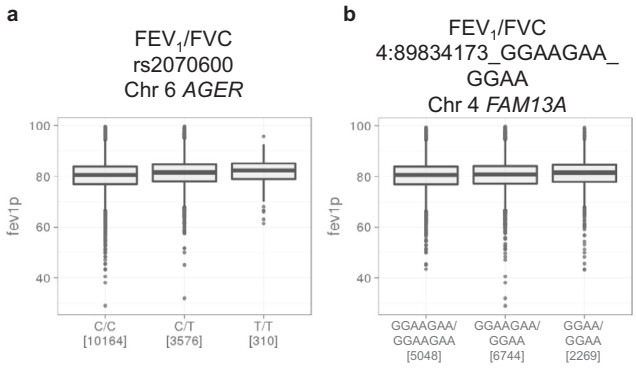

**a** FEV₁/FVC rs2070600 Chr 6 AGER

**b** FEV₁/FVC 4:89834173_GGAAGAA_GGAA Chr 4 FAM13A

**Fig. 4 The effects of genetic variants repeated at both Discovery and Validation Stage for FEV₁/FVC. a, b** The measures of FEV₁/FVC in each genotype of the sentinel variant are shown. Boxes represent the inter-quartile range (IQR) between first quartile (Q1) and third quartile (Q3), the line inside represents the median. Whiskers denote the lowest and highest values within $1.5 \times$ IQR from the Q1 and Q3 respectively. Black dots represent outliers beyond the whiskers. FEV₁ forced expiratory volume in one second, FVC forced vital capacity.

variants; 603 SNPs plus 93 insertion/deletions) with $p < 5 \times 10^{-8}$ for FeNO (Fig. 6b, Table 2, Supplementary Data 4). The sentinel variants of eleven loci for FeNO were located at 17q11.2 in *NOS2~LYRM9*, 6p22.3 in *C6orf62~GMNN*, 12p13.31 in *DSTNP2*, 13q22.1 in *KLF5~LINC00392*, 18q21.2 in *TCF4*, 11q23.3 in *TREH~DDX6*, 3p25.1 in *LSM3~LINC01267*, 5q31.1 in *C5orf66*, 22q11.21 in *TBX1*, 8p21.2 in *SLC25A37~NKX3-1*, and 5q13.3 in *PDE8B*.

**Genetic variants for FeNO repeated at both Discovery and Validation Stage.** To validate potential associations between FeNO and genetic loci identified at the Discovery Stage, we then conducted genome-wide analysis in the participants of the ToMMo BirThree Cohort as Validation Stage (Fig. 1, Supplementary Fig. 9a). Of the eleven loci identified at the Discovery Stage for FeNO, we found two loci conformed in tier 1; 17q11.2 in *NOS2~LYRM9* and 12p13.31 in *DSTNP2* had an association $p < 5 \times 10^{-8}$ both at the Discovery Stage and Validation Stage with consistent directions of effect (Table 2, Supplementary Fig. 9b, Supplementary Data 5). We also found that 6p22.3 in *C6orf62~GMNN* had an association $p < 5 \times 10^{-5}$ with consistent directions of effect at the Validation Stage and conformed in tier 2 (Table 2, Supplementary Fig. 9b, Supplementary Data 5). Detailed examinations of these associations have been performed (Figs. 7, 8).

The strongest association with FeNO at both stages was at 17q11.2 with the top variant, an intergenic SNP rs2531870 between *NOS2* and *LYRM9* (Table 2). The regional association plot (Fig. 7a) suggested that the probable candidate gene at this locus is *NOS2*, which codes inducible NO synthase (iNOS)[42]. The protective minor T allele of SNP rs2531870 was associated with lower FeNO ($p = 1.2 \times 10^{-29}$), with the lowest mean FeNO among TT homozygotes at the Discovery Stage (Fig. 8a).

Another tier 1 locus that showed $p < 5 \times 10^{-8}$ association with FeNO at both stages was at 12p13.31 with the sentinel SNP rs11064445 in *DSTNP2* (Table 2). Whereas *DSTNP2* is a pseudogene that is not likely to affect FeNO, the regional association plot (Fig. 7b) identified a plausible candidate gene in the proximity of this locus that was related to FeNO, *SPSB2* (Supplementary Data 5). *SPSB2* encodes a SPRY domain-containing SOCS (suppressor of cytokine signalling) box protein 2 (SPSB2). It has been reported that SPSB2 is a negative regulator that recruits an E3 ubiquitin ligase complex and induces polyubiquitination of iNOS, resulting in proteasomal

degradation[43]. The protective allele A of the sentinel SNP rs11064445 at 12p13.31 was associated with lower FeNO ($p = 6.2 \times 10^{-17}$), with the lowest mean FeNO among AA homozygotes at the Discovery Stage (Fig. 8b).

We also found 6p22.3 with the sentinel SNP rs76800089 in *C6orf62~GMNN* as a tier 2 locus for FeNO. The regional association plot (Fig. 7c) identified that the plausible candidate gene at this locus related to FeNO was *RIPOR2* (Supplementary Data 5). *RIPOR2* encodes Rho Family-Interacting Cell Polarization Regulator 2, also referred to as FAM65B. RIPOR2 negatively regulates the migration, adhesion and proliferation of T cells[44,45], which are the cells responsible for allergic inflammation including the induction of iNOS in airway epithelial cells by their cytokines. The aggravated minor T allele of rs76800089 was associated with higher FeNO ($p = 1.3 \times 10^{-27}$), with the highest mean FeNO among TT homozygotes at the Discovery Stage (Fig. 8c).

We also performed a series of GWAS for FeNO with the adjustment by the principal components (PC1~10). The results including significantly associated loci were not significantly different between the GWAS with and without the adjustment by the principal components (Supplementary Figs. 10 and 11).

To further explore the function of these identified three loci for FeNO, we again utilized the GTEx Analysis V8 Fine-Mapping cis-eQTL data of lung tissue and whole blood to find out the genes that expressions are related to the haplotypes of identified genetic variants in the three loci. To our expectations, the analyses revealed that the genetic variants suggested by our study are related to the expression of identified genes or candidates that we have mentioned (17q11.2 for *NOS2*, 2p13.31 for *SPSB2*, 6p22.3 for *RIPOR2*; Supplementary Fig. 12), suggesting that the identified genetic variants affect the FeNO level through the expression of *NOS2*, *SPSB2*, and *RIPOR2*.

In summary, we conducted GWAS exploiting two-independent populations and identified three genetic loci associated with the FeNO levels in adults. The proteins encoded by the candidate genes identified in our study are iNOS, SPSB2, and RIPOR2, all of which have functions that could influence the levels of FeNO and the phenotypes of obstructive airway diseases including asthma and COPD.

**Genetic correlation of FeNO with asthma-related traits.** Because FeNO is the biomarker for type 2 airway inflammation and the measurement of FeNO is widely used for a diagnosis of asthma, we examined the genetic correlation between FeNO and asthma using LD score regression (LDSC). To this end, we utilized the BioBank Japan GWAS resource as the traits including the risk of asthma in the Japanese subjects[46–48] (Supplementary Fig. 13a, Supplementary Table 6). When we analysed genetic correlation with the risk of COPD, two allergic diseases (pollinosis and atopic dermatitis), and quantitative traits related to asthma (blood eosinophil count and body mass index (BMI)), we found significant correlations of FeNO with blood eosinophil count ($r_g = 0.37$, $p = 0.002$) after Bonferroni correction for the number of pairwise comparisons. In contrast, while pollinosis showed nominal ($p < 0.05$) significance with FeNO ($r_g = 0.52$, $p = 0.03$), the disease did not show significant correlation after Bonferroni correction. The risk of asthma did not show significant correlation with FeNO, although there was a tendency of positive correlation between FeNO and asthma ($r_g = 0.16$, $p = 0.13$).

In addition to the genetic correlation analyses, we also examined whether the 11 loci that showed association with FeNO were overlapped with asthma, other allergic diseases, blood eosinophil count, or lung function from the NHGRI-EBI GWAS Catalog[49] (Supplementary Fig. 13b). Importantly, two out of 11 loci were found to be overlapped with asthma. Two other loci were associated with allergic rhinitis and/or allergic dermatitis.

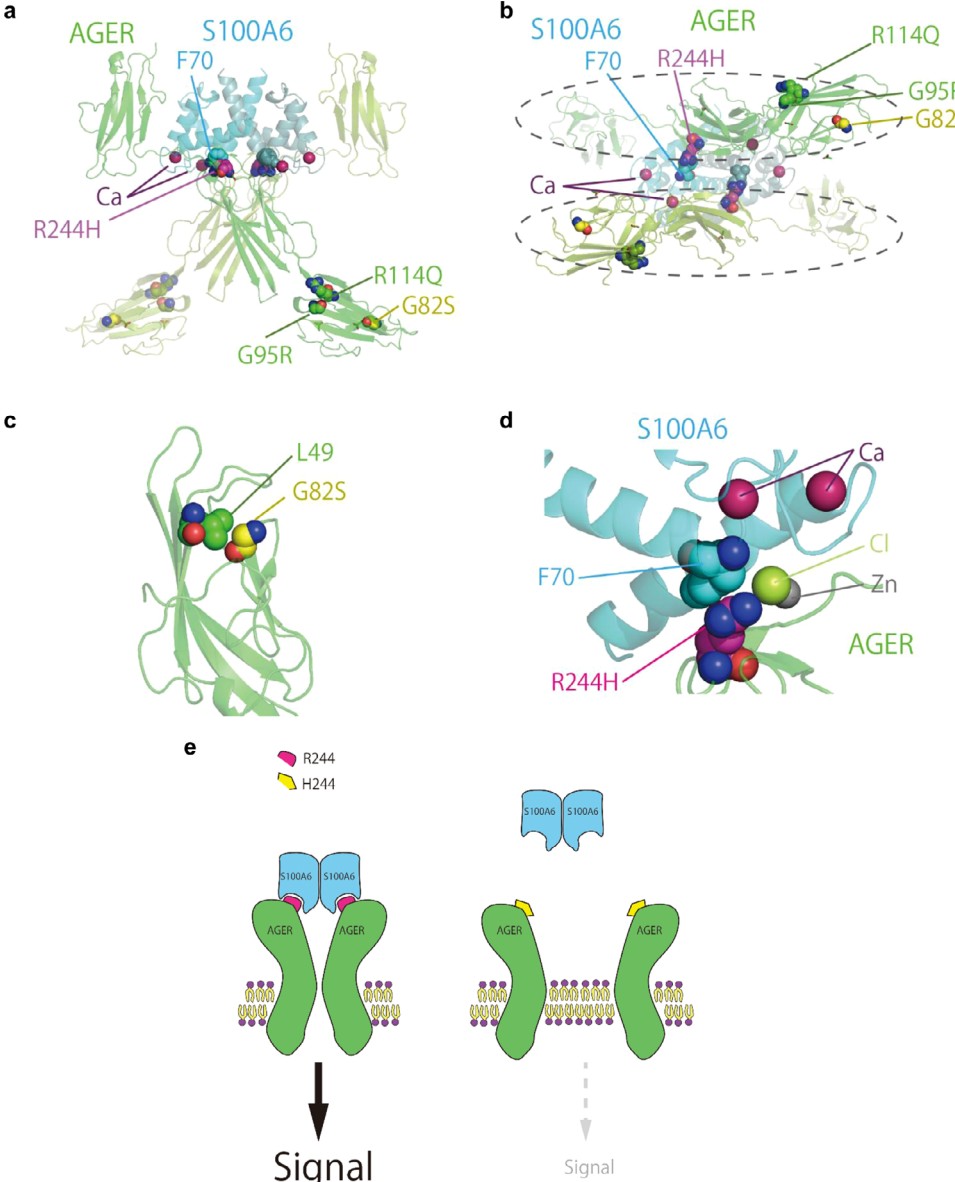

**Fig. 5 The presumable effects of missense variants of AGER suggested by the analysis of three-dimensional structure. a** Side view of the AGER (light and dark green) and S100A6 (light and dark cyan) heterotetramer. **b** The same tetramer as in (**a**), viewed from the axis of symmetry. The dashed ellipses show two AGER domains, which are not in direct contact with each other. **c** Close up view of the G82 (yellow) and L49 (green) of AGER. **d** Close up view of the AGER R244 (magenta) and S100A6 F70 (cyan) residues. **e** The explanatory scheme of the effect of the R244H substitution on the signalling of AGER. The R244H substitution can impair the interaction between a AGER and a S100A6 because two AGER subunits are in indirect contact with each other via the S100A6 dimer, which results in the inhibition of the dimerization and signalling of the AGER.

Four loci were associated with the blood eosinophil count, the biomarker for allergic diseases. For lung function, two loci were associated with $FEV_1/FVC$ and one locus associated with $FEV_1$. These analyses support our contention that there are genetic links between FeNO and asthma-related traits, and there exist common genetic background for allergic diseases and biomarkers for allergy including FeNO.

## Discussion

We performed GWAS for the lung function utilizing two strategically established adult cohort populations to identify genetic vulnerabilities for respiratory diseases including COPD. This is the first large-scale GWAS study for lung function in Japan. Our Discovery Study identified the *AGER* locus at 6p21.32 including exonic nonsynonymous SNP rs2070600 in *AGER* as the locus

most significantly associated with $FEV_1/FVC$ in the Japanese population. This locus was nicely replicated in the Validation Stage. The 3D structure-based assessment suggests that the SNP in *AGER* introduces a G82S substitution into one of the immunoglobulin domains in AGER, which likely affects the molecular function of the receptor. Moreover, based on the relationship between common variants in GWAS and rare variants in the genome reference panel, we performed an extensive search of our genomic allele reference panel, 4.7KJPN[39], and identified three rare missense variants of AGER. Of note, R244H appears to be critical for the dimerization and signalling of the AGER. We also performed GWAS for FeNO and identified three genetic loci associated with the FeNO levels in the adult populations. These loci are all replicated in the Validation Stage. The proteins encoded by the candidate genes in the loci are *iNOS*, *SPSB2*, and

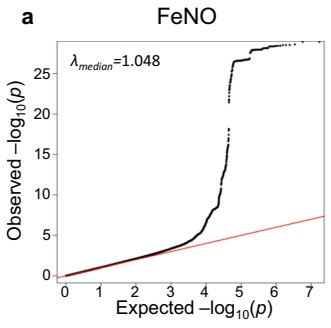

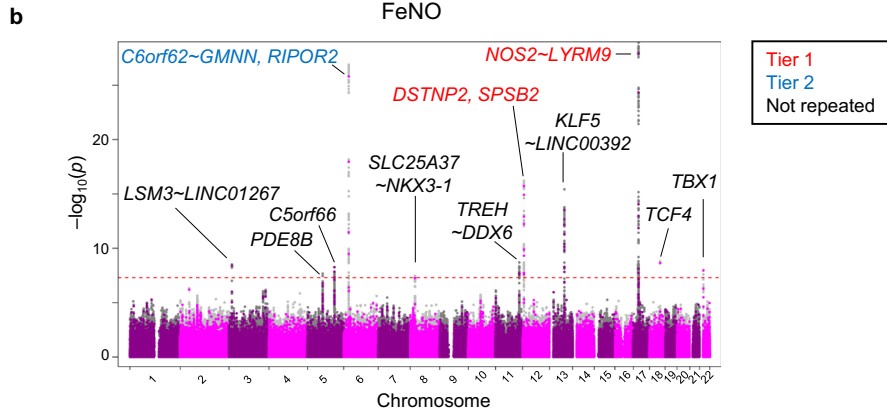

**Fig. 6 GWAS for association with the levels of FeNO at the Discovery Stage. a** A Quantile-quantile (QQ) plot of the observed versus the expected log10(P) values in FeNO is shown with the inflation factor (λ). **b** The Manhattan plot of GWAS for the association with FeNO shows the chromosomal position of variants exceeding the genome-wide significance threshold ($P < 5 \times 10^{-8}$ as indicated by the dotted red line). Gene names in red correspond to tier 1 loci ($P < 5 \times 10^{-8}$ both in the Discovery Stage and Validation Stage); gene names in blue correspond to tier 2 loci ($P < 5 \times 10^{-8}$ in the Discovery Stage and $P < 5 \times 10^{-5}$ in the Validation Stage with consistent directions of effect); gene names in black correspond to genetic loci which had $P < 5 \times 10^{-8}$ in Discovery Stage but not repeated by Validation Stage. Coloured circles mean the variants directly analysed. Grey circles mean the variants detected by genotype imputation. GWAS genome-wide association study, FeNO exhaled nitric oxide.

*RIPOR2*, which have functions that influence the levels of FeNO. Thus, our study provides an insight into the understanding of the pathogenesis of respiratory diseases including COPD and airway type 2 inflammation.

Our GWAS revealed that the genetic locus including *AGER* had the strongest association with FEV₁/FVC. AGER is a pattern recognition receptor that binds to a variety of damage-associated molecular pattern molecules including advanced glycation end products (AGEs), high mobility group protein B1 (HMGB1), and S100 proteins[36]. The interaction between AGER and the ligands is considered to induce inflammatory responses by the activation of nuclear factor kappa B (NF-κB)[50,51]. On note, recent larger-scale meta-analysis of the European ancestry GWAS for FEV₁/FVC revealed that the *p*-value of the *AGER* locus is the lowest one[20], suggesting that loss of AGER function may affect FEV₁/FVC similarly in individuals both of European and Japanese ancestry.

At this 6p21.32 locus, we observed two top variants, an exonic nonsynonymous SNP rs2070600 in AGER and intronic insertion/deletion rs41315238 in RNF5 at the Discovery Stage. It has been well reported that rs2070600 in *AGER* is associated with FEV₁/FVC measures[13,20,52,53] as well as the risk of COPD[19]. In contrast, to the best of our knowledge, the association of *RNF5* rs41315238 with FEV₁/FVC has not been identified in previous GWAS for FEV₁/FVC and the risk of COPD. However, the linkage disequilibrium analysis suggests that *RNF5* rs41315238 is in near complete linkage disequilibrium with *AGER* rs2070600. Therefore, our observation also suggests that the locus 6p21.32

including these sentinel SNPs are important for lung function as previously reported[13,20,52,53].

Our GWAS showed that the minor T allele of nonsynonymous exonic SNP rs2070600 in *AGER* is protective and associated with higher FEV₁/FVC, suggesting that this nonsynonymous variant is functional. Three-dimensional structure analysis of AGER also suggests that a G82S substitution induced by the rs2070600 affects the molecular function of AGER. Since our present GWAS study clearly identified that AGER is an influential molecule for lung function in the Japanese population, we further explored the presence of other nonsynonymous variants of AGER that influence molecular and biological functions of the protein in the Japanese population using our genomic allele frequency panel, 4.7KJPN[39]. We consider that those common variants in GWAS to rare variants in the whole-genome reference panel approach provide important information. Indeed, we have identified three rare missense variants of AGER. The protein structure analysis further revealed that one of the variants, R244H, is likely to inhibit the dimerization and signalling of AGER, because the structural analysis clearly indicated that this mutation affects the interaction between AGER and its ligand, S100A6. Functional and biological analyses of this variant in the laboratory as well as in COPD patients remain to reveal the precise mechanism of the R244H substitution in the pathogenesis of COPD.

The *AGER* loci containing rs2070600 is located in the human leukocyte antigen (HLA) or major histocompatibility complex (MHC) region. The MHC region contains both HLA genes and the genes related to inflammation and immune responses. The

**Table 2 Loci associated with the values of FeNO.**

| Pheno-type | Locus | Gene | Sentinel SNP | Function | Ref | Alt | EA | Impu-ted | Discovery stage | | | | Validation stage | | | | Verified | Rep-orted |
|---|---|---|---|---|---|---|---|---|---|---|---|---|---|---|---|---|---|---|
| | | | | | | | | | EA freq. | Beta | SE | P | EA freq. | Beta | SE | P | | |
| FeNO | 17q11.2 | NOS2-LYRM9 | rs2531870 | Inter-genic | T | C | C | Y | 0.7281 | 0.1437 | 0.0129 | 1.2E-29 | 0.7161 | 0.1590 | 0.0207 | 4.1E-14 | Tier 1 | Y |
| | 6p22.3 | C6orf62-GMNN | rs76800089 | Inter-genic | C | T | T | Y | 0.0610 | 0.2583 | 0.0242 | 1.3E-27 | 0.0598 | 0.1602 | 0.0389 | 4.4E-05 | Tier 2 | N |
| | 12p13.31 | DSTNP2 | rs11064445 | ncRNA_exonic | G | A | A | Y | 0.2028 | -0.1177 | 0.0142 | 6.2E-17 | 0.2045 | -0.1219 | 0.0228 | 5.3E-08 | Tier 1 | N |
| | 13q22.1 | KLF5-LINC00392 | rs9573166 | Inter-genic | G | A | A | Y | 0.4626 | -0.0931 | 0.015 | 3.6E-16 | 0.4648 | -0.0675 | 0.0183 | 1.8E-04 | N | N |
| | 18q21.2 | TCF4 | rs8090085 | intronic | T | G | G | Y | 0.3695 | 0.0721 | 0.0120 | 1.6E-09 | 0.3661 | 0.0732 | 0.0189 | 1.4E-04 | N | N |
| | 11q23.3 | TREH-DDX6 | rs10892277 | Inter-genic | G | A | A | Y | 0.2064 | -0.0849 | 0.0143 | 3.0E-09 | 0.2003 | 0.0025 | 0.0233 | 9.9E-01 | N | N |
| | 3p25.1 | LSM3-LINC01267 | rs4685105 | Inter-genic | A | G | G | Y | 0.4247 | 0.0656 | 0.016 | 2.4E-09 | 0.4249 | 0.0074 | 0.0186 | 6.0E-01 | N | N |
| | 5q31.1 | C5orf66 | rs6891074 | ncRNA_intronic | C | T | T | N | 0.5891 | 0.0670 | 0.017 | 5.5E-09 | 0.5945 | 0.0276 | 0.0188 | 1.3E-01 | N | N |
| | 22q11.21 | TBX1 | rs737869 | intronic | G | C | C | Y | 0.5450 | 0.0675 | 0.016 | 1.0E-08 | 0.5415 | 0.0421 | 0.0186 | 2.9E-01 | N | N |
| | 5q13.3 | PDE8B | rs7702192 | intronic | C | A | A | Y | 0.4381 | -0.0630 | 0.016 | 2.1E-08 | 0.4353 | -0.0319 | 0.0186 | 1.1E-01 | N | N |
| | 8p21.2 | SLC25A37-NKX3-1 | rs11988717 | Inter-genic | G | A | A | Y | 0.3984 | 0.0663 | 0.018 | 3.4E-08 | 0.3915 | 0.0352 | 0.0189 | 5.2E-02 | N | N |

Ref reference allele, Alt alternative allele, EA effect allele, EA freq. EA frequency, SE standard error, ncRNA non-coding RNA, UTR untranslated region.

*AGER* is located in the MHC class III region that does not harbour genes involved in antigen presentation. However, because it is between class I and class II regions and all MHC genes possess haplotype transmission, it is possible that the variants in *AGER* could be in linkage disequilibrium (LD) in some MHC haplotypes that might be related to the phenotypes including lung function and the risk of COPD.

The GWAS to identify susceptibility loci for adult asthma in the Japanese population have revealed that the SNP rs404860 in the MHC class III region has a significant association with the risk of adult asthma and this SNP shows relatively weak LD with rs2070600[54]. The study for AGER G82S polymorphism introduced by rs2070600 on rheumatoid arthritis (RA), a representative autoimmune and inflammatory disease, showed that G82S showed the association with the risk of RA. However, after correction for the presence/absence of HLA DRB1*0401, this association was lost, indicating that rs2070600 is not associated with RA independently of HLA DRB1*[55].

To our best knowledge, there was no direct or strong evidence that shows the pathogenic linkage between rs2070600 and MHC haplotype in terms of the $FEV_1$/FVC or the risk of COPD. We rather surmise that following studies may be pertinent. Investigation on the relation between plasma AGER levels and *AGER* genetic variants in COPD patients revealed that rs2070600 is associated with circulating sAGER levels[56]. *AGER* rs2070600 polymorphism influenced the expression of both AGER mRNA and protein[57]. These studies suggest that the rs2070600 polymorphism influences the expression and function of AGER, directly influencing the lung function and the risk of COPD, but *AGER* rs2070600 polymorphism does not influence lung function and COPD through indirect effects in terms of the linkage with MHC haplotypes.

We performed GWAS for lung function with the adjustment by FeNO level, because we wished to identify genetic factors related to lung function irrespective to the existence of type 2 inflammation. However, the result of GWAS for $FEV_1$/FVC with FeNO adjustment did not show significant difference from that without the adjustment. The reason why FeNO, an indicator of type 2 inflammation, did not influence the GWAS for $FEV_1$/FVC is unclear at present. One plausible explanation is that the type 2 airway inflammation is reversible and may not be strongly correlated to the airway obstruction in steady state.

Our GWAS study identified three genetic loci (at 17q11.2, 12p13.31 and 6p22.3) associated with the levels of FeNO in adult populations. A GWAS study for FeNO in children has been reported[58]. It is interesting to note that the genetic locus at 17q11.2 is also identified as an associated locus for FeNO, whereas the other two loci are identified in this study are not identified in the study. Similarly, our GWAS did not find significant association of 17q21.1 with FeNO, which is identified in the previous children GWAS[58]. We surmise that the difference between children and adults or the difference of ethnicity may serve as reasons for this inconsistency. Alternatively, the difference may be attributable to the sample size, as the sample size of our Discovery GWAS is much larger than that of the paediatric GWAS.

Both the regional association plot and the cis-eQTL analyses suggest that the probable candidate genes at these loci are *NOS2*, *SPSB2*, and *RIPOR2*. Importantly, the functions of the three proteins encoded by these genes are clearly involved in the production of nitric oxide. Firstly, *NOS2* encodes iNOS, which is an enzyme responsible for exhaled NO production in normal and asthmatic subjects[59]. IL-4 and IL-13, which are produced by inflammatory cells including type 2 helper T cells during allergic inflammation including asthma, are known to induce the expression of iNOS in airway epithelial cells[60,61]. Secondly, SPSB2

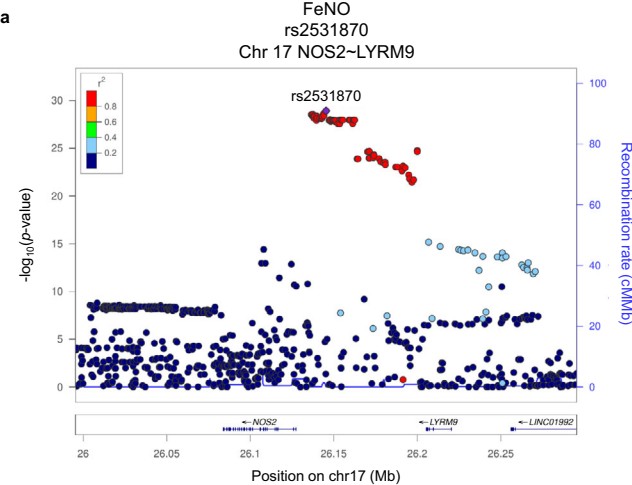

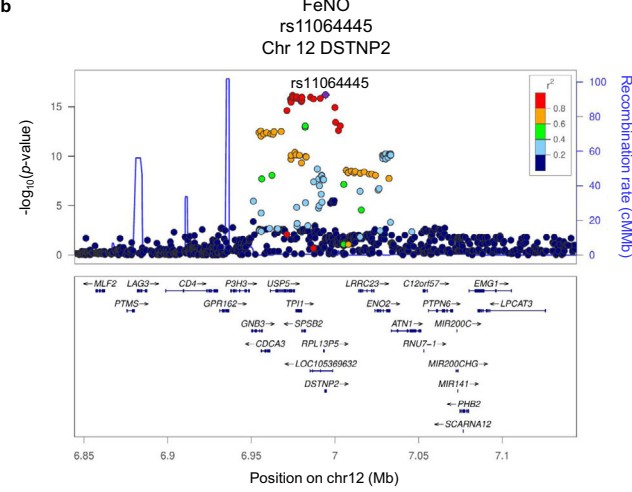

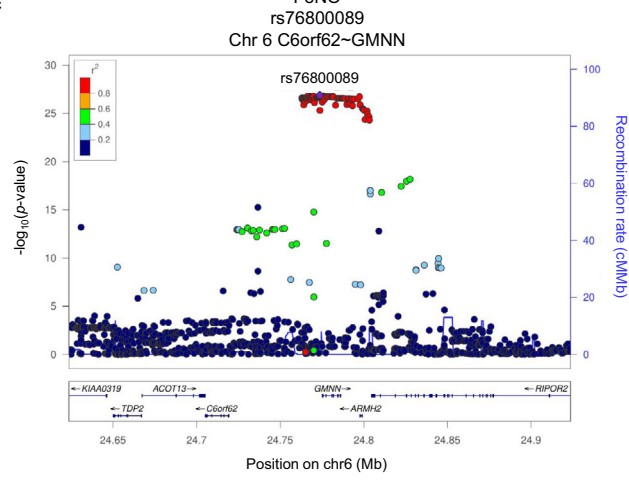

**Fig. 7 Regional association plots of the loci associated with FeNO at both Discovery and Validation stages. a–c** Statistical significance of each genetic variant on the −log10 scale as a function of chromosome position (hg19) in the Discovery stage alone. The sentinel variant at each locus is shown in purple. The correlations ($r^2$) of each of the surrounding variants to the sentinel variant are shown in the indicated colours. The loci shown in (**a**, **b**) are tier 1 locus which showed an association with FeNO with $P < 5 \times 10^{-8}$ both in the Discovery Stage and Validation Stage. **c** is tier 2 locus which showed an association with FeNO with $P < 5 \times 10^{-8}$ in the Discovery Stage and $P < 5 \times 10^{-5}$ in the Validation Stage with consistent directions of effect. FeNO exhaled nitric oxide.

iNOS degradation. This clearly shows that this genetic variant upregulates the expression of a negative regulator of iNOS, which reduces the level of FeNO in human subjects. Thirdly, RIPOR2 is also called FAM65B, which negatively regulates the activation and proliferation of effector T cells[44,45], which produce the cytokines that induce iNOS in airway epithelial cells. It has been well documented that increased production of NO causes nitrosative stress, nitrosylation of redox-sensitive thiols, and is intimately involved in the pathogenesis of respiratory diseases including COPD and asthma[62–65]. Taken together, we propose that the genetic variants identified in these three genes influence the phenotypes of respiratory diseases including COPD and asthma.

To explore whether there exists a common genetic background between asthma and FeNO, a biomarker for type 2 airway inflammation and widely used for the diagnosis of asthma, we further performed analyses of genetic correlation between FeNO and the risk of asthma as well as the traits related to asthma. The analyses showed the tendency of a positive correlation between FeNO and the risk of asthma. We surmise that the lack of significant correlation between FeNO and asthma in this analysis reflects the limitation of the sample size. This hypothesis is further supported by the finding that there are significant correlations between FeNO and blood eosinophil count. Moreover, two loci out of the 11 loci that showed the association with FeNO in our GWAS are found to be overlapped with the risk of asthma. Alternative explanation for the observation is to assume possible difference of genetic background between FeNO and the risk of asthma, but we interpret the latter as remote.

Finally, we must discuss the limitations in our study. First, the size of the population, especially for the Validation Stage, is relatively small, which may have caused a lower repeat ratio of the genetic loci in this study. We realize that we need to recruit more participants. Nonetheless, it is important to note that we have identified a significant number of repeated loci, including, to the best of our knowledge, newly identified loci for FeNO. Second, we need to confirm whether the identified genetic variants actually alter the functions and/or expressions of the gene products. We are planning fine expression analyses as well as loss-of-function analyses for these variants. Especially, the generation of a mouse line with an AGER rare variant mutation is now ongoing. In this regard, we would like to note that these genetic variants are significantly associated with the measures of lung function or FeNO, and the functions of target genes are reported to be involved in the pathogenesis of emphysema or the synthesis of NO[43–45,66,67].

In summary, our large-scale GWAS study with Discovery and Validation Stages has identified the SNP rs2070600 in *AGER* as the most influential locus associated with $FEV_1/FVC$ in the adult population in Japan. The three-dimensional protein structure analysis of AGER revealed that the SNP is likely to affect the molecular function of the receptor. Further analyses of *AGER* using our genomic allele reference panel identified three other rare missense variants of AGER and one of them, R244H, appears

is an adopter protein that recruits an E3 ubiquitin ligase complex and induces polyubiquitination and degradation of iNOS[43], so that SPSB2 works as a negative regulator for the production of NO. Importantly, cis-eQTL analysis has revealed that the protective allele A of the sentinel SNP rs11064445 at 12p13.31, which is associated with lower FeNO in our GWAS, is significantly related to higher expression of mRNA for SPSB2 that accelerates

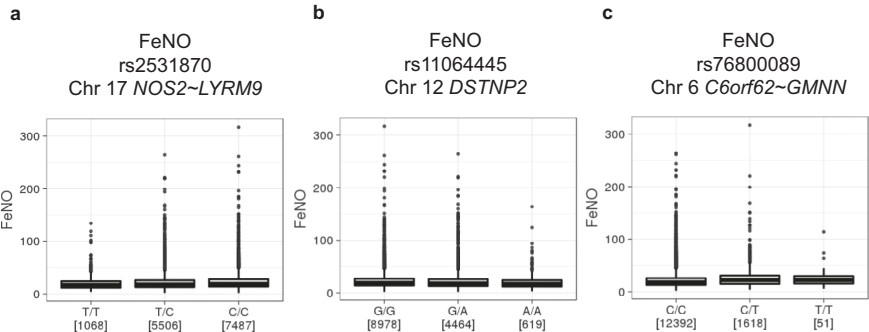

**Fig. 8 The effects of genetic variants repeated at both Discovery and Validation Stage for FeNO. a–c** The levels of FeNO in each genotype of the sentinel variant are shown. Format of the boxplots is the same as in Fig. 4. FeNO exhaled nitric oxide.

to be critical for the dimerization and signalling of AGER. We also identified three genetic loci associated with the FeNO levels. The candidate genes in these loci are *NOS2*, *SPSB2* and *RIPOR2*, which are all involved in the production of NO. We believe that these findings will contribute to a better understanding of the pathogenesis of respiratory diseases, especially COPD and asthma. Further investigations in the laboratory as well as in patients could elucidate the pathogenic role of the genes and their variants identified in this study.

## Methods

**Study design and population**. All the studies related to the TMM project were approved by the ethics committee of the Tohoku University School of Medicine and the ethics committee of the ToMMo in Tohoku University. Participants in this study were recruited from the Miyagi prefecture, from 1 June 2017 until 2 July 2019. Written informed consent was obtained from each participant when they were enroled in the cohort studies.

In our GWAS for lung function and FeNO, we used $FEV_1$ measures, $FEV_1/FVC$ measures, FeNO levels, phenotype information for covariates, and the genotype data of 14,061 Japanese individuals from ToMMo CommCohort Study for the Discovery Stage and those for 5661 Japanese individuals from ToMMo BirThree Cohort Study as an independent dataset for the Validation Stage. The $FEV_1$, $FEV_1/FVC$, and FeNO levels were measured in the medical examination for the cohort participants.

**Lung Function Test and FeNO measurement**. We measured the respiratory function parameters including $FEV_1$ and FVC using HI-801 (Chest M.I. Inc., Tokyo, Japan). All test results were regularly checked by trained physicians in a blind manner, and invalid data were excluded from the data set. Similarly, we measured FeNO using NIOX VERO (Circassia AB, UK). Participants were measured for both respiratory functions and FeNO. We further checked the collected data to determine whether there were outliers or systematic differences among the measured assessment centres or measured seasons.

**Genotyping, imputation, and quality control**. To prepare direct and imputed genotype datasets, cohort participants were genotyped with Affymetrix Axiom Japonica Array (v2) separately in 19 batches. We set the number of plates for each batch to 50 according to the guidelines for Axiom array data genotyping[68]. The Japonica Array (v2) analysis including genotype imputation were conducted as described[69] with IMPUTE2 (ver. 2.3.2)[70] using a phased reference panel of 3552 Japanese individuals from cohort studies of the TMM project[71]. For quality control, we excluded plates with an average call rate <0.95 and removed samples with DishQC metric <0.82 or step1 call rate <0.97 before batch genotyping. After batch genotyping, we removed samples with a call rate <0.95, or having unusually high IBD values with many other samples. We also excluded variants with a *p*-value of the Hardy-Weinberg equilibrium (HWE) test <$1.00 \times 10^{-5}$, minor allele frequency (MAF) < 0.01, or missing rate >0.01 in each batch. We merged the imputed genotype datasets for the 19 batches using QCTOOL (v2.0.4) (https://www.well.ox.ac.uk/gav/qctool/) with the following options: -omit-chromosome -compare-variants-by position,allele -flip-to-match-cohort1. As a result, we obtained an imputed genotype dataset in Oxford BGEN format for 90,565 Japanese individuals with 54,034,112 variants. We also obtained a direct genotype dataset in PLINK BED format for the 90,565 individuals with 659,326 variants by merging the genotype datasets before the imputation for the 19 batches.

From our Japonica Array analysis, we extracted 14,115 individuals who were from CommCohort Study and were without any missing information of the phenotype and covariate values considered in our GWAS for the Discovery Stage. We also extracted 5672 individuals who were from the BirThree Cohort Study and

were without any missing information of the phenotype and covariate values, for the Validation Stage. We excluded 47 individuals from the Discovery Stage and nine individuals from the Validation Stage due to discrepancies in reported gender. We also excluded seven individuals from the Discovery Stage and two individuals from the Validation Stage to avoid pairs of individuals with pi-hat value >0.75. We calculated pi-hat values for the exclusion using the direct genotype dataset and PLINK (v1.90b5.1) with the following options:-maf 0.05, -hwe 0.05 -geno 0.01 -indep-pairwise 1500 150 0.03 -genome. We again applied an SNP quality control step to the imputed and direct genotype datasets for the Discovery and Validation Stages to exclude variants with *p*-value of HWE test <$1.00 \times 10^{-5}$, MAF < 0.01, or missing rate >0.05 for imputed and >0.01 for direct genotype, and IMPUTE2 info score <0.8 for imputed dataset. The number of remaining variants in each dataset for GWAS is 8,587,571 in the Discovery Stage and 8,595,665 in the Validation Stage.

**Genome-wide association analysis**. We used BOLT-LMM v2.3.2[72] for linear-mixed model analysis in order to test the additive genetic effect of SNPs on the $FEV_1$, $FEV_1/FVC$, and FeNO level for autosomal chromosomes. In our GWAS analysis, biases from the population stratification as well as familial and cryptic relatedness were controlled by considering the genetic correlation matrix in the linear-mixed model[73]. To reduce the skewness and kurtosis of the distribution of $FEV_1$, $FEV_1/FVC$, and FeNO, Box-Cox transformation was applied to these data using the R package (car ver.2.1.5; https://cran.r-project.org/web/packages/car/citation.html). The following information was used as covariates for the adjustment: (i) age, sex, height, and smoking status for $FEV_1$; (ii) age, sex, smoking status, and FeNO level for $FEV_1/FVC$; and (iii) age, height, and smoking status for the FeNO level. Required information for BOLT-LMM other than the genotype and phenotype data were: LDscoresFile: East Asian LD Scores (https://data.broadinstitute.org/alkesgroup/LDSCORE/eas_ldscores.tar.bz2), geneticMapFile: genetic_map_hg19.txt.gz; and modelSnps: text files including the information of 214,837 and 215,346 SNPs that exist both in the file for East Asian LD Scores and the non-imputed genotype dataset for the Discovery and Validation Stages, respectively. In the GWAS for each phenotype, we gave the direct genotype dataset with -bfile option and the imputed genotype dataset with -bgenFile option to BOLT-LMM. We merged the results for the direct genotype dataset and the results for the imputed genotype dataset by overwriting the latter results with former results for variants existing in both of them.

We utilized *p*-value of $5 \times 10^{-8}$ as the genome-wide significance level to identify the genetic locus associated with the trait at Discovery Stage GWAS. To validate potential associations between the trait and the genetic locus suggested at Discovery Stage, we utilized both *p*-value of $5 \times 10^{-8}$ and *p*-value of $5 \times 10^{-5}$ as the genome-wide suggestive level at Validation Stage GWAS of which size of participants is smaller than that of Discovery Stage. Based on these two-stage analyses, we set up three tiers according to the certainty. Tier 1 loci had $p < 5 \times 10^{-8}$ at both Discovery Stage and Validation Stage. Tier 2 loci had $p < 5 \times 10^{-8}$ at the Discovery Stage and $5 \times 10^{-8} \leq p < 5 \times 10^{-5}$ at the Validation Stage with consistent directions of effect. Tier 3 loci had $p < 5 \times 10^{-8}$ at Discovery Stage but their *p* was more than $5 \times 10^{-5}$ at the Validation Stage.

**Genetic correlation analysis**. We calculated genetic correlations for all the pair of $FEV_1/FVC$, $FEV_1$, FeNO, and the following traits by applying LDSC[74] to their GWAS summary statistics: asthma, COPD, pollinosis, atopic dermatitis, drug eruption, eosinophil count, and BMI. GWAS summary statistics of $FEV_1/FVC$, $FEV_1$, and FeNO were obtained from our GWAS results, while those of the remaining traits in Japanese population were obtained from the BioBank Japan GWAS resource[46–48] (https://humandbs.biosciencedbc.jp/en/hum0014-v21). Note that, for obtaining the GWAS summary statistics of $FEV_1/FVC$ and $FEV_1$ for LDSC, FeNO was not included in the covariates. As the LD scores, which was also required in LDSC, we used East Asian LD Scores from 1000 Genomes Project (https://data.broadinstitute.org/alkesgroup/LDSCORE/eas_ldscores.tar.bz2). We

excluded variants that were not included in the East Asian LD Scores, with MAF < 0.01, or with the imputation quality score (INFO metric or Rsq) <0.9, from the calculation of LDSC.

**Identifying causal variants from GWAS and eQTL**. We explored functions of the identified signals in intergenic regions by the detection of genes whose expression changes were associated with those signals based on the expression quantitative trait locus (eQTL) data. We used the Genotype-Tissue Expression (GTEx) V8 Analysis dataset as the eQTL data, which contains eQTL results for 54 types of tissues and is available in GTEx Portal (https://gtexportal.org/home/). We selected 'Lung' and 'Whole Blood' as the relevant tissues for our GWAS traits among the 54 types of tissues, and limited the use of the eQTL results for those two tissues. We selected the highly confident eQTL associated variants for each gene from the significantly associated eQTL variants based on the posterior probabilities of causal variant statuses estimated by CAVIAR[75], a method for identifying causal variants from GWAS and eQTL results. The posterior probabilities by CAVIAR were pre-computed in the GTEx V8 Analysis dataset, and we set the threshold of the posterior probabilities for the extraction to 0.05. For each signal in an intergenic region, we then selected genes for which there existed at least one highly confident eQTL associated variant located within 1 kb distance from one of genomewide-significant variants comprising the signal and identified them as the genes whose expression changes were possibly caused by the signal.

**Linkage disequilibrium analysis**. We calculated two linkage disequilibrium LD measures, $r^2$ and D', using PLINK 1.90b5.1 for variants in chr6:32120000–32180000, a neighbouring region of rs10947233, rs41315238, and rs2070600. These LD measures were calculated from the genotype data of Discovery Stage.

**Reporting summary**. Further information on research design is available in the Nature Research Reporting Summary linked to this article.

## Data availability

The GWAS summary statistics generated in this study are available via GWAS Catalog (https://www.ebi.ac.uk/gwas/downloads/summary-statistics) under study accession identifiers GCST90044841, GCST90044842, GCST90044843, GCST90044844, GCST900 44845 and GCST90044846. Summary GWAS statistics are also available at the Japanese Multi Omics Reference Panel (jMorp) website (https://jmorp.megabank.tohoku.ac.jp/gwas/; ID TGA000010).

Individual genotyping results and other cohort data used for the association study are stored in Tohoku Medical Megabank Organization. In response to reasonable requests for these data (contact us at dist@megabank.tohoku.ac.jp), we will share the stored data after assembling the data set after approval of the Ethics Committee and the Materials and Information Distribution Review Committee of Tohoku Medical Megabank Organization.

Source data underlying box plots shown in figures are provided in Supplementary Data 6.

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

## Acknowledgements

We thank Mr. Brent K. Bell for critical reading of the manuscript and language assistance. We thank all participants and all municipality staffs who helped our project. We also thank the members of ToMMo for their assistance to the projects. We thank Bio-Bank Japan for the GWAS resource to analyse the genetic correlations of the trait of our GWAS study with other traits including the risk of asthma.

The Genotype-Tissue Expression (GTEx) Project was supported by the Common Fund of the Office of the Director of the National Institutes of Health, and by NCI, NHGRI, NHLBI, NIDA, NIMH, and NINDS. The data used for the analyses described in this manuscript were obtained from the GTEx Portal.

This work was supported by grants from the Japan Agency for Medical Research and Development (AMED); AMED Advanced Genome Research and Bioinformatics Study to Facilitate Medical Innovation (GRIFIN) project [grant number JP20km0405203] and the Tohoku Medical Megabank Project from the Ministry of Education, Culture, Sports, Science, and Technology (MEXT) of Japan and AMED [grant numbers JP20km0105001 and JP20km0105002]. All computational resources were provided by the ToMMo supercomputer system (http://sc.megabank.tohoku.ac.jp/en), which is supported by the Facilitation of R&D Platform for AMED Genome Medicine Support conducted by AMED [grant number JP20km0405001]. This work was supported partially by Platform Project for Supporting Drug Discovery and Life Science Research [Basis for Supporting Innovative Drug Discovery and Life Science Research (BINDS)] from AMED under Grant # JP19am0101067 to K.K.

## Author contributions

M. Yamada, I.M., N. Fuse., and M. Yamamoto conceived the study. N. Fuse., A.H., S.K., T.N., S.O., Y.H, K.S., J.S., A.U., and E.K. collected and managed the ToMMo sample and information. I.M., K. Kinoshita, F.K., S.T. and K. Kojima performed genotyping and whole-genome sequencing. M. Shirota performed 3D structure analysis of the AGER complex. I.M., K. Kojima and M. Sakurai performed statistical analysis. N. Fujino, T.N., T.I., A.M., and T.O. analysed the data of lung function and FeNO. K. Kinoshita, F.K., M.I., and H.S. supervised the study. M. Yamada, I.M., K. Kojima, M. Shirota, and M. Yamamoto wrote the manuscript.

## Competing interests

The authors declare no competing interests.
