## [Transparent Peer Review File · Communications Biology]

Reviewers' comments:

Reviewer #1 (Remarks to the Author):

In this study, Dr. Yamada et al. performed GWASs of FEV1/FVC, FEV1, and FeNO. Several genetic loci have been identified and replicated. Some of my concerns are:

Majors:

- (1) I have lots of questions related to covariates adjusted in the model:
 - (1a) It's not very convincing to adjust FeNO levels for GWAS of FEV1/FVC. Correlation between FeNO and FEV1/FVC is not significant ($p=0.14$) in this study.
 - (1b) If authors believe adjustment of FeNO is essential for their findings, please also run GWAS without adjustment of FeNO and make comparison to prove their assumption.
 - (1c) FeNO levels is adjusted for GWAS of FEV1/FVC only, but not GWAS of FEV1. Does it mean authors believe FeNO is not correlated with FEV1?
 - (1d) Height is adjusted for GWAS of FEV1 only, but not FEV1/FVC, why?
 - (1e) Gender is not adjusted for GWAS of FeNO, why?
- (2) How was potential relatedness between subjects adjusted? Were related subjects removed, especially Bir Three Cohort which looks like a family study? I saw $\pi\text{-hat}>0.75$, which is very relaxed. I did not see PCA included as covariates in the model.
- (3) The Results section is too wordy. Lots of information such as gene functions and previous findings should go to the Discussion section, instead of repeating in Results and Discussion.
- (4) Authors should spend more effort on novel findings. AGER (rs2070600) has been identified to be associated with FEV1, FEV1/FVC, and COPD many times. NOS2 has also been identified to be associated with FeNO levels before (van der Valk, JACI, 2013). Authors stated this is the first large GWAS of FeNO, which may not be true.
- (5) GTEx eQTL database may be surveyed to indicate potential function of SNPs identified in this study.

Minors:

- (1) There are some typos in the manuscript. For example, P15, "form" should be "from"; P24, "FCV" should be "FVC".
- (2) Table 1, it might be better to include %asthma and %COPD.
- (3) Table S6, it's not clear what is FEV1%.

Reviewer #2 (Remarks to the Author):

This article contributed with novel findings on the genetics of lung health and COPD, since is the first large-scale genome-wide analysis based solely on Japanese individuals. The authors also brought novelty to this area of research by undertaking a GWAS analysis using the fractional exhaled nitric oxide (FeNO) levels as the phenotype of question, which is an indicator of inflammation (and asthma).

The GWAS analyses were carried out in two independent cohort studies of the Northeast coast of Japan, using the CommCohort ($N\sim 14K$) for a Discovery stage and BirThree ($N\sim 5K$) for a Validation stage. This two-stage GWAS strategy was conducted to test for association with spirometry measures FEV1 and FEV1/FVC, and to test for association with FeNO levels. The authors used a well-established 3-Tier-system to define associated lung function signals, being Tier 1 signals the most significant (reaching $P < 5 \times 10^{-8}$ at both Discovery and Validation stages).

The most significant associated signal with lung function was the nonsynonymous variant rs2070600 in AGER (6p21.32). Although this signal has been previously reported to be significantly associated with FEV1/FVC, I found it nicely of the authors going one step further into mapping the interaction

effects of rs2070600 to the three-dimensional structure of the AGER complex. Interestingly, they found that this variant introduces a G82S substitution that may explain the effect alteration and its association with lung function. The authors also investigated the three-dimensional protein interaction of three additional missense rare variants of AGER that were found in their allele reference allele, providing a richer functional understanding. It would be worth investigating whether conducting a GWAS without FeNO levels as covariate would also brought up this signal.

The authors first identified signals associated with FeNO levels, these included two Tier 1 variants in candidate genes NOS2~LYRM9 and DSTNP2, and a Tier 2 signal in RIPOR2, all of which are most likely implicated in the pathophysiology of asthma. I did not find particularly meaningful the genetic correlation analysis between FeNO levels and asthma (and other related traits) as it was expected to see a positive rg.

In general, I found the topic of this article original and the manuscript well-written. The chosen methodology and statistics is adequate, and the content is in agreement with the abstract, and the discussion part also strengthens the main findings. The biggest limitation of this manuscript, as the authors mentioned in the discussion is the small sample size. A suggestion for a continuing article, if not possible to incorporate into the current one is to be able to combine individuals from the used cohorts to The BioBank Japan. Moreover, the reader might find interesting if there were a further exploration of the defined lung function signals in comparison to other ancestries. The approach given in Koyama, Satoshi, et al. "Population-specific and trans-ancestry genome-wide analyses identify distinct and shared genetic risk loci for coronary artery disease." *Nature genetics* 52.11 (2020): 1169-1177, might be suitable for future research bringing a much in-depth knowledge of lung function and COPD of the Japanese population.

Introduction

- Page 4 – line 21. Provide a stronger justification of why you decided to conduct lung function GWAS on Japanese population. How unified is the Japanese genetics, and how diverse it is from other East Asian populations where similar studies have already been performed.

- Page 4 – line 24: The author states that there are significantly different characteristics between Japanese COPD patients compared to Europeans, however it is not provided to the reader an overview of this (e.g. emphysema vs chronic bronchitis or decreased tobacco sensitivity in Japan?). Additionally, how different/similar are the Japanese COPD patients with respect with other studies based on East Asian population.

- Page 5. The author mentions that The Tohoku Medical Megabank (TMM) is comprised by two prefectures; Miyagi that operates ToMMO and Iwate, which operates the Iwate TMM Organization. It is not clear whether the two cohort studies are based from both prefectures. Moreover, on line 22 it is stated that this research was conducted in the two independent cohorts of ToMMO. The question then is whether CommCohort Study and BirThree Cohort Study are only from the Miyagi prefecture, but not from Iwate? This is not specified.

Results

- From both GWAS strategies (with and without FeNO adjustment) is there any tendency of effect sizes from the lead variants that have been previously reported compared to other ancestries, e.g., the effect size were generally larger in the Japanese study in relation to European studies, or are these comparable? What about the allele frequencies?

- Page 9 – line 22. Worth considering performing a control genome-wide association study using individuals from The BioBank Japan (2,774 COPD cases in BBJ; PMID: 28189464) without FeNO adjustment.

- Page 9 – line 18. What is the MAF of rs2070600 in this study, and how does it compares to other populations (EUR, EAS)? For example, what was the MAF of this signal in the SpiroMeta study?
- It is not mentioned that rs2070600 is located in the human leukocyte antigen (HLA) region and further implications.
- Page 15 - line 5 and 9. $P = 0.03$ and $P = 0.14$ are not significant.

Methods

- No mention whether chromosome X was included or excluded from the analysis.
- What was the chosen imputation method (reference panel and software)? The paper cited is not yet published and is not included in the reference list. Please provide a brief description of the imputation strategy. Additionally, did you apply any filter for low quality imputed variants ($R^2 \geq 0.3$) before conducting the GWAS analysis? This is only mentioned in the genetic correlation analysis methods.
- Was there any genomic inflation?
- Was the Box-Cox transformation the reason that it was omitted to add age2 as part of your covariates? Please provide a justification.
- Page 23 – line 15. Reference needed for used R package as provided here: <https://cran.r-project.org/web/packages/car/citation.html>

Figures and Tables

- It would be helpful for the reader to include the minor allele frequencies (MAF) percentage of all signals in the supplementary tables.

Typos:

- P7.11 and 13. Space inserted between number and percentage symbol.
- P-value significance threshold is sometimes written with a space in between and sometimes without the space. Suggest to be consistent.
- Page 17 – line 24. GAWS -> GWAS
- Page 24 -line 13. Node -> Note

Reviewer #3 (Remarks to the Author):

This is a nicely thought out and implemented study that addresses the great need on non-Caucasian genomic studies in respiratory disease. The authors utilise two large Japanese cohorts as discovery and validation cohorts which provides a strong case for their observations.

I recommend that this manuscript is considered for publication on addressing the following concerns:

Major Concerns:

1. Page 7 line 21 - I am unsure as to why the authors have normalised FEV1/FVC to age and gender. The value is a ratio and is therefore independent of gender and age, unlike FEV1. This means that there may be over correction in the analysis which can skew results
2. Page 10 line 5: The authors mention that the observed SNP is a novel reported association for FEV1/FVC. This is not accurate. The reported SNP is in near complete Linkage Disequilibrium ($R^2: 0.97$) in the 1000 genomes East Asian Population with rs2070600, which indicates that both SNPs tag the same association locus. rs2070600 has been previously reported as a locus of association for FEV1/FVC (PMID: 20010834, 30804560, 28166213, 23284291) and COPD (PMID: 30804561). This must be discussed in the manuscript, with comparisons of B-value to the authors own results
3. Page 12, line 6 - Another large scale FeNo GWAS has been carried out albeit in children (PMID: 24315451), using a mixed population. This study needs to be discussed in relation to the authors'

study

4. There is a general lack of discussion with relation to haplotypes and related findings. The manuscript will greatly benefit from further in silico analysis to relate the authors' finding to the wealth of genetic information out there, especially eQTL and pQTL data to provide confidence on the selected genes for the GWAS loci.

5. In the discussion the authors claim that in Caucasian studies the AGER locus was not a 'top hit' unlike in the Japanese population. I am unsure what this refers to. Is this in relation to P-value or B-value? Comparisons must consider the B-value with regards magnitude of association. Are B-values in all Caucasian studies really all smaller than in this study?

Minor Concerns:

1. Page 4 line 22 - could the authors expand on the statement that the Japanese COPD population is different to the Caucasian one?

2. A p-value of 5×10^{-8} and 5×10^{-5} are reported. I assume these values are corrected for multiple testing and are different due to different population numbers. This needs to be clarified in text

3. Table 1: Median values for population data are more informative than Mean values. Can these be changed?

Responses to the comments from the reviewers

Reviewer 1:

Comments to the Author:

In this study, Dr. Yamada et al. performed GWASs of FEV₁/FVC, FEV₁, and FeNO. Several genetic loci have been identified and replicated.

Response: We thank the reviewer for providing valuable comments. We have addressed the reviewer's concerns in point-by-point responses, with each comment and response numbered sequentially as C and R, respectively.

Majors:

C1-Major-1: I have lots of questions related to covariates adjusted in the model.

C1-Major-1a: It's not very convincing to adjust FeNO levels for GWAS of FEV₁/FVC. Correlation between FeNO and FEV₁/FVC is not significant (p=0.14) in this study.

C1-Major-1b: If authors believe adjustment of FeNO is essential for their findings, please also run GWAS without adjustment of FeNO and make comparison to prove their assumption.

R1-Major-1a and 1b: We thank the reviewer for the professional advice about adjustment with FeNO for FEV₁/FVC GWAS. Because we wished to perform the GWAS for FEV₁/FVC, a lung function measure for evaluating airway obstruction, with adjustment by an indicator of type 2 airway inflammation that can influence lung function measures, especially that for airway obstruction, we utilized FeNO for the adjustment of FEV₁/FVC GWAS. Additional reason is that multiple regression analysis in our cohorts suggested that there is a significant and independent correlation between FEV₁/FVC and FeNO (please see reviewer only data 1).

We agree with the reviewer's advice, and therefore, in this revision we have conducted additional GWAS for FEV₁/FVC with or without FeNO adjustment. This analysis gives rise to an important discovery. To our surprise, the result of GWAS with

FeNO adjustment is similar to that without the adjustment, although the *P*-value of genetic locus 4q22.1 in *FAM13A* is lower in the GWAS with FeNO adjustment than that without adjustment (please see Supplementary Fig.1). Following the result of reperformed GWAS without FeNO adjustment, we modified the results section and deleted the sentences mentioning about the effect of FeNO adjustment as a possible reason for the differences between our GWAS results and those of the previous world-wide multicentre GWAS.

C1-Major-1c: FeNO levels is adjusted for GWAS of FEV₁/FVC only, but not GWAS of FEV₁. Does it mean authors believe FeNO is not correlated with FEV₁.

R1-Major-1c: We did not adjust the GWAS for FEV₁ with FeNO because one previous study has reported that FeNO is not related to FEV₁¹. However, following the reviewer's comment, we additionally performed the GWAS of FEV₁ with the adjustment by FeNO. The result was not significantly different from that without the adjustment with FeNO. Therefore, we keep the results of GWAS of FEV₁ without the adjustment by FeNO in this manuscript.

C1-Major-1d: Height is adjusted for GWAS of FEV₁ only, but not FEV₁/FVC, why?

R1-Major-1d: As reviewer 3 commented in the C3-1, FEV₁/FVC is a ratio that is not significantly influenced by physical constitution and its parameter, like height. In fact, a previous report showed that both age and gender are independently associated with FEV₁/FVC, but height is not². In contrast, FEV₁ has been shown to be significantly influenced by physical constitution and its parameter, like height². Based on these reasons, we have used height for the adjustment of GWAS for FEV₁ only, but not for FEV₁/FVC.

C1-Major-1e: Gender is not adjusted for GWAS of FeNO, why?

R1-Major-1e: As previous reports studying about FeNO suggested that gender is not a significantly influencing factor^{1,3}, we did not use gender for the adjustment of GWAS

for FeNO. We agree with the reviewer's comment, and additionally performed the GWAS of FeNO with the adjustment by gender. The results were not significantly different from those of the GWAS without the adjustment by gender. We have attached the results as reviewer only data 2. Therefore, we keep the results of GWAS of FeNO without the adjustment by gender in this manuscript.

C1-Major-2: How was potential relatedness between subjects adjusted? Were related subjects removed, especially Bir Three Cohort which looks like a family study? I saw $\pi\text{-hat} > 0.75$, which is very relaxed. I did not see PCA included as covariates in the model.

R1-Major-2: We thank the reviewer for these comments. In fact, the BirThree cohort is a family study. This time we did not remove related subjects in our GWAS, as in our GWAS analysis, biases from the population stratification as well as familial and cryptic relatedness were controlled by considering the genetic correlation matrix in the linear-mixed model⁴. We have added the description to the Method.

According to the reviewer's suggestion, we reperformed our GWAS with the adjustment by the principal components (PC1~10). However, the results including significant associated loci were not significantly different between the GWAS with and without the adjustment by the principal components. Please see attached reviewer only data 3 to 8. Therefore, we keep the results of GWAS without the adjustment by the principal components in this manuscript.

C1-Major-3: The Results section is too wordy. Lots of information such as gene functions and previous findings should go to the Discussion section, instead of repeating in Results and Discussion.

R1-Major-3: We appreciate the reviewer's suggestion. Following the suggestion, we have modified the Results and condensed the information of genes function and previous findings.

C1-Major-4: Authors should spend more effort on novel findings. *AGER* (rs2070600) has been identified to be associated with FEV₁, FEV₁/FVC, and COPD many times. *NOS2* has also been identified to be associated with FeNO levels before (van der Valk, JACI, 2013). Authors stated this is the first large GWAS of FeNO, which may not be true.

R1-Major-4: We thank the reviewer for these important comments. We agree with the comments. We have revised the text and incorporated the information on *AGER* and *NOS2* as recommended. As for the FeNO GWAS, we apologize that our description was not articulate. There was a GWAS for FeNO for approximately 8,900 children, but those for adults were at best for approximately 1,200. Therefore, we believe that our GWAS for FeNO (approximately 14,000 for discovery plus 5,600 for replicate) is the largest-scale GWAS for adults in terms of the number of subjects.

C1-Major-5: GTEx eQTL database may be surveyed to indicate potential function of SNPs identified in this study.

R1-Major-5: This is a precious comment, and we agree. Indeed, we have worked on the analyses. We have utilized GTEx Portal, a public database provided by The Genotype-Tissue Expression (GTEx) project (<https://www.gtexportal.org/home/>) to find out the genes, which expression are related to the haplotypes of genetic variants suggested by our GWAS. The analyses revealed that the genetic variants suggested by our study is related to the expression of genes including the candidate genes that we mentioned in the manuscript (by FEV₁/FVC GWAS: 6p21.32 for *AGER* and *RNF5*, 4q22.1 for *FAM13A*; by FeNO GWAS: 17q11.2 for *NOS2*, 2p13.31 for *SPSB2*, 6p22.3 for *RIPOR2*). These results are added to Supplementary Fig.3 and Fig.5, and described in the text. These results reinforce our findings of GWAS, as the identified genetic variants affect the traits (FEV₁/FVC or FeNO) through the expression of candidate genes. Once again, we thank the reviewer for this precious comment.

C1-Minor-1: There are some typos in the manuscript. For example, P15, "form"

should be "from"; P24, "FCV" should be "FVC".

R1-Minor-1: We apologize for these oversights. We have corrected the typos throughout the text, including those suggested by the reviewer.

C1-Minor-2: Table 1, it might be better to include %asthma and %COPD.

R1-Minor-2: We have revised Table 1 and added the information of %asthma and %COPD.

C1-Minor-3: Table S6, it's not clear what is FEV1%.

R1-Minor-3: We apologize for confusion. FEV₁/FVC should be the correct word.

Reviewer 2:

Comments to the Author:

In general, I found the topic of this article original and the manuscript well-written. The chosen methodology and statistics is adequate, and the content is in agreement with the abstract, and the discussion part also strengthens the main findings. The biggest limitation of this manuscript, as the authors mentioned in the discussion is the small sample size. A suggestion for a continuing article, if not possible to incorporate into the current one is to be able to combine individuals from the used cohorts to The BioBank Japan. Moreover, the reader might find interesting if there were a further exploration of the defined lung function signals in comparison to other ancestries. The approach given in Koyama, Satoshi, et al. might be suitable for future research bringing a much in-depth knowledge of lung function and COPD of the Japanese population.

Response: We thank the reviewer for constructive and professional comments, which helped us to improve our manuscript. We have addressed the reviewer's concerns in point-by-point responses, with each comment and response numbered sequentially as C and R, respectively. We also thank the reviewer for thoughtful suggestions about

overcoming the small sample-size limitation of our study utilizing the data of the BioBank Japan, about exploration of the defined lung function signals by our study in comparison to other ancestries, and about our future study to bring a much in-depth knowledge of lung function and COPD of the Japanese population.

Introduction

C2-1: Page 4 – line 21. Provide a stronger justification of why you decided to conduct lung function GWAS on Japanese population. How unified is the Japanese genetics, and how diverse it is from other East Asian populations where similar studies have already been performed.

R2-1: We thank the reviewer for this important comment. The main reason to perform the lung-function GWAS on Japanese population is that a recent whole-genome sequence data in the Northeast Asian Reference Database (NARD) revealed that the ancestral composition of Japanese is different from those of other East Asian populations including Korean and Han Chinese⁵. This information leads us the notion that it is worth performing GWAS of lung function in Japanese, as Japanese are different from the other East Asians in terms of ancestral compositions. Besides, the sizes of lung-function GWAS in East Asia were rather limited, at best less than 8,900, we wish to perform larger scale GWAS than those reported previously^{6,7}. We appreciate the reviewer to raise this point. We have modified the Introduction and incorporated these aspects.

C2-2: Page 4 – line 24: The author states that there are significantly different characteristics between Japanese COPD patients compared to Europeans, however it is not provided to the reader an overview of this (e.g. emphysema vs chronic bronchitis or decreased tobacco sensitivity in Japan?). Additionally, how different/similar are the Japanese COPD patients with respect with other studies based on East Asian population.

R2-2: We agree with the reviewer about the importance to describe clinical

characteristics of Japanese COPD patients. We have added the information that the proportion of patients having emphysema-dominant phenotype is greater in Japanese COPD population than those in the US and Europe COPD population^{8,9}. We also added a sentence mentioning that the frequency of exacerbation in Japanese COPD population is smaller than those in the US and Europe COPD populations¹⁰. As the reviewer suggested, characteristics of COPD patients in Korea are similar to those in Japanese COPD patients in terms of emphysema-dominant phenotype and less frequent exacerbators¹¹⁻¹³. We have incorporated these observations into the Introduction.

C2-3: Page 5. The author mentions that The Tohoku Medical Megabank (TMM) is comprised by two prefectures; Miyagi that operates ToMMO and Iwate, which operates the Iwate TMM Organization. It is not clear whether the two cohort studies are based from both prefectures. Moreover, on line 22 it is stated that this research was conducted in the two independent cohorts of ToMMo. The question then is whether CommCohort Study and BirThree Cohort Study are only from the Miyagi prefecture, but not from Iwate? This is not specified.

R2-3: We apologize for the confusion mentioning about TMM and ToMMo. The cohorts utilized in this manuscript are only from the Miyagi prefecture. We revised the sentences in Introduction and Methods.

Results

C2-4: From both GWAS strategies (with and without FeNO adjustment) is there any tendency of effect sizes from the lead variants that have been previously reported compared to other ancestries, e.g., the effect size were generally larger in the Japanese study in relation to European studies, or are these comparable? What about the allele frequencies?

R2-4: We would like to ask the reviewer to understand that the exact comparison of the effect size of the lead variants between our study and European studies is difficult,

because we applied Box-Cox transformation to FEV₁, FEV₁/FCV, and FeNO to reduce the skewness and kurtosis of the distribution of these traits. The method for reducing the skewness and kurtosis in European studies is not completely the same one that we have utilized.

As for the minor allele frequency, we can compare. For example. minor allele frequency of rs2070600 is larger in our study than that in European studies. Please see the response to comment C2-6.

C2-5: Page 9 – line 22. Worth considering performing a control genome-wide association study using individuals from The BioBank Japan (2,774 COPD cases in BBJ; PMID: 28189464) without FeNO adjustment.

R2-5: We thank the reviewer for this comment about the effect of adjustment by the levels of FeNO to the results of GWAS for FEV₁/FVC. Reviewer 1 also advised us similar comment. Therefore, in this revision we have performed GWAS for FEV₁/FVC again with or without FeNO adjustment.

To our surprize, the result of GWAS with FeNO adjustment is similar to that without the adjustment, although the *P*-value of genetic locus 4q22.1 in *FAM13A* is lower in the GWAS with FeNO adjustment than that without adjustment (please see Supplementary Fig.1). Therefore, we have deleted the sentences speculating the effect of FeNO adjustment as a possible reason for that our GWAS is different from the previous world-wide multicentre GWAS.

C2-6: Page 9 – line 18. What is the MAF of rs2070600 in this study, and how does it compares to other populations (EUR, EAS)? For example, what was the MAF of this signal in the SpiroMeta study?

R2-6: MAF of rs2070600 is 0.1493 in both Discovery and Validation cohorts (Table 1). The MAF of rs2070600 is 0.0639 in the meta-analysis of UK Biobank and SpiroMeta¹⁴, both of which studies European ancestry individuals were participated. We have also investigated the public database (gnomAD¹⁵ and NARD⁵) and summarized the MAF of

rs2070600 in each ethnic population and added Supplementary Table 7 to the revised manuscript. The MAF is larger in East Asian population than the other ethnic populations, including European ancestry populations. Of the East Asian populations, the MAF of Han Chinese seems to be larger than Japanese and Korean populations.

C2-7: It is not mentioned that rs2070600 is located in the human leukocyte antigen (HLA) region and further implications.

R2-7: We thank the reviewer suggesting that rs2070600 and its gene *AGER* is located in the human leukocyte antigen (HLA) or major histocompatibility complex (MHC) region. The MHC region contains both HLA genes and the genes related to inflammation and immune responses. The *AGER* is located in the MHC class III region that does not harbour genes involved in antigen presentation. However, because it is between class I and class II regions and all MHC genes possess haplotype transmission, it is possible that the variants in *AGER* could be in linkage disequilibrium (LD) in some MHC haplotypes that might be related to the phenotypes including lung function and the risk of COPD.

The GWAS to identify susceptibility loci for adult asthma in the Japanese population have revealed that the SNP rs404860 in the MHC class III region has significant association with the risk of adult asthma and this SNP shows relatively weak LD with rs2070600¹⁶. The study for *AGER* G82S polymorphism introduced by rs2070600 on rheumatoid arthritis (RA), a representative autoimmune and inflammatory disease, showed that G82S showed the association with the risk of RA. However, after correction for the presence/absence of HLA DRB1*0401, this association was lost, indicating that rs2070600 is not associated with RA independently of HLA DRB1*¹⁷.

To our best knowledge, there was no direct or strong evidence that shows the pathogenic linkage between rs2070600 and MHC haplotype in terms of the FEV₁/FVC or the risk of COPD. We rather surmise that following studies may be pertinent. Investigation on the relation between plasma *AGER* levels and *AGER* genetic variants

in COPD patients revealed that rs2070600 is associated with circulating sRAGE levels¹⁸. *AGER* rs2070600 polymorphism influenced the expression of both *AGER* mRNA and protein¹⁹. These studies suggest that the rs2070600 polymorphism influences the expression and function of *AGER*, directly influencing the lung function and the risk of COPD, but *AGER* rs2070600 polymorphism does not influence lung function and COPD through indirect effects in terms of the linkage with MHC haplotypes. We described this argument in Discussion.

C2-8: Page 15 - line 5 and 9. P = 0.03 and P = 0.14 are not significant.

R2-8: We thank the reviewer for the suggestion about *P*-value significance threshold for genetic correlation analysis. As the reviewer pointed out, only blood eosinophil count showed the significant correlation with FeNO ($r_g = 0.37$, $P = 0.002$) after Bonferroni correction for the number of pairwise comparisons. Polynosis only showed the nominal ($P < 0.05$) significance with FeNO ($r_g = 0.52$, $P = 0.03$), but did not have the significant correlation after Bonferroni correction. We have revised the sentence mentioning that the risk of asthma did not have the significant correlation with FeNO, although there was a tendency of positive correlation between these two traits ($r_g = 0.16$, $P = 0.13$). We also have revised the sentences in Results and Discussion, and Fig. 9 and Supplemental Table 6 for the genetic correlation analyses.

Methods

C2-9: No mention whether chromosome X was included or excluded from the analysis.

R2-9: In our GWAS study, we studied only autosomal genes, but chromosome X was excluded. We have mentioned this in Methods.

C2-10: What was the chosen imputation method (reference panel and software)? The paper cited is not yet published and is not included in the reference list. Please provide a brief description of the imputation strategy. Additionally, did

you apply any filter for low quality imputed variants ($R^2 \geq 0.3$) before conducting the GWAS analysis? This is only mentioned in the genetic correlation analysis methods.

R2-10: We apologize for lack of explanation about the imputation method. A paper describing the method was recently accepted for publication by *J Biochem*²⁰. We have added succinct description of the imputation strategy to the Method, citing the accepted paper. As for the reviewer's question, we have reperformed our GWAS applying the filter of IMPUTE2 info score < 0.8 for imputed dataset.

C2-11: Was there any genomic inflation?

R2-11: We have added the inflation factor (λ) to the Q-Q plots.

C2-12: Was the Box-Cox transformation the reason that it was omitted to add age² as part of your covariates? Please provide a justification.

R2-12: We wished to answer this comment experimentally, in this revision we reperformed the GWAS analysis with the Box-Cox transformation of age. We found that the GWAS results with the Box-Cox transformation of age were not significantly different from those without the Box-Cox transformation of age. Therefore, we have kept the GWAS results without the Box-Cox transformation of age in this manuscript.

C2-13: Page 23 – line 15. Reference needed for used R package as provided here: <https://cran.r-project.org/web/packages/car/citation.html>

R2-13: We have added the reference for R package to the text.

Figures and Tables

C2-14: It would be helpful for the reader to include the minor allele frequencies (MAF) percentage of all signals in the supplementary tables.

R2-14: We thank the reviewer for this professional comment. We have added the information of each effect allele and its frequency for each signal to the supplementary

tables.

Typos:

C2-15: P7.11 and 13. Space inserted between number and percentage symbol.

R2-15: We apologize for these oversights. We have deleted space between number and percentage symbol.

C2-16: P-value significance threshold is sometimes written with a space in between and sometimes without the space. Suggest to be consistent.

R2-16: We have corrected to be that *P*-value significance threshold should be consistently written with a space in between.

C2-17: Page 17 – line 24. GAWS -> GWAS

R2-17: We have corrected this typo.

C2-18: Page 24 -line 13. Node -> Note

R2-18: We have corrected this typo.

Reviewer 3:

Comments to the Author:

This is a nicely thought out and implemented study that addresses the great need on non-Caucasian genomic studies in respiratory disease. The authors utilise two large Japanese cohorts as discovery and validation cohorts which provides a strong case for their observations.

We would like to thank the reviewer for constructive and helpful comments. We have addressed the reviewer's concerns extensively in this revision.

Major Concerns:

C3-1: Page 7 line 21 - I am unsure as to why the authors have normalised FEV₁/FVC to age and gender. The value is a ratio and is therefore independent of gender and age, unlike FEV₁. This means that there may be over correction in the analysis which can skew results.

R3-1: We thank the reviewer for the professional and important comment. We would like to inform the reviewer that there is a report showing that both age and gender are independently associated factors with FEV₁/FVC². Indeed, in our cohorts, multiple linear regression analysis revealed that both age and gender, as well as smoking status and FeNO, are independently related factors to FEV₁/FVC (please see reviewer only data 1). Since we are seeking the genetic background that affects FEV₁/FVC independently of age and gender, we adjusted both age and gender for our GWAS of FEV₁/FVC.

C3-2: Page 10 line 5: The authors mention that the observed SNP is a novel reported association for FEV₁/FVC. This is not accurate. The reported SNP is in near complete Linkage Disequilibrium (R²: 0.97) in the 1000 genomes East Asian Population with rs2070600, which indicates that both SNPs tag the same association locus. rs2070600 has been previously reported as a locus of association for FEV₁/FVC (PMID: 20010834, 30804560, 28166213, 23284291) and COPD (PMID: 30804561). This must be discussed in the manuscript, with comparisons of B-value to the authors own results.

R3-2: We thank the reviewer for this important comment. As the reviewer pointed out, rs41315238 in *RNF5* is in near complete linkage disequilibrium in our GWAS (Fig.3b). Therefore, we have deleted and/or corrected the sentences in Results and Figures (Fig.3a and Fig.4b) mentioning the *RNF5* and its SNP.

C3-3: Page 12, line 6 - Another large scale FeNo GWAS has been carried out albeit in children (PMID: 24315451), using a mixed population. This study needs

to be discussed in relation to the authors' study.

R3-3: We agree with this comment and deleted the sentence “To our best knowledge, this is the first large scale GWAS trial with FeNO” in Results.

As the reviewer pointed out, a large scale GWAS (N = 8, 858) for FeNO using 14 independent paediatric cohorts was performed and identified 3 SNPs associated with FeNO values: rs3751972, rs944722 and rs8069176. The rs3751972 in LYR motif containing 9 (*LYRM9*; $P = 1.97 \times 10^{-10}$) and rs944722 in inducible nitric oxide synthase 2 (*NOS2*; $P = 1.28 \times 10^{-9}$) are both located at 17q11.2, while rs8069176 near gasdermin B (*GSDMB*; $P = 1.88 \times 10^{-8}$) locus is located at 17q21.1. An intriguing observation is that top hit SNP of our GWAS rs2531870 is located in the intergenic region between *NOS2* and *LYRM9* at 17q11.2, suggesting that this genetic locus is associated with FeNO both in children and adults. However, we could not find significant associations in a genetic locus at 17q21.1. Conversely, the paediatric GWAS did not find the association between FeNO and genetic loci at 6p22.3 and 12p13.31, both of which our GWAS identified associations with FeNO in two independent cohorts reproducibly. We surmise that the difference between children and adults or the difference of ethnicity may serve as reasons for this inconsistency. Alternatively, the difference may be attributable to the sample size, as the sample size of our Discovery GWAS is much larger than that of the paediatric GWAS. We have modified Discussion and incorporated succinctly a discussion summarized above, that is the findings revealed in the comparison between the paediatric GWAS and our adult GWAS for FeNO.

C3-4: There is a general lack of discussion with relation to haplotypes and related findings. The manuscript will greatly benefit from further in silico analysis to relate the authors' finding to the wealth of genetic information out there, especially eQTL and pQTL data to provide confidence on the selected genes for the GWAS loci.

R3-4: We sincerely appreciate the reviewer's precious comment. Indeed, we have

worked on the analyses. We have utilized GTEx Portal, a public database provided by The Genotype-Tissue Expression (GTEx) project (<https://www.gtexportal.org/home/>) to find out the genes, which expression are related to the haplotypes of genetic variants suggested by our GWAS. The analyses revealed that the genetic variants suggested by our study is related to the expression of genes including the candidate genes that we mentioned in the manuscript (by FEV₁/FVC GWAS: 6p21.32 for *AGER* and *RNF5*, 4q22.1 for *FAM13A*; by FeNO GWAS: 17q11.2 for *NOS2*, 2p13.31 for *SPSB2*, 6p22.3 for *RIPOR2*). These results are added to Supplementary Fig.3 and Fig.5 and described in the text. These results reinforce our findings of GWAS, as the identified genetic variants affect the traits (FEV₁/FVC or FeNO) through the expression of candidate genes.

C3-5: In the discussion the authors claim that in Caucasian studies the *AGER* locus was not a 'top hit' unlike in the Japanese population. I am unsure what this refers to. Is this in relation to P-value or B-value? Comparisons must consider the B-value with regards magnitude of association. Are B-values in all Caucasian studies really all smaller than in this study?

R3-5: We agree with your comment and apologize for the lack of following up literature study. Although the previous paper of the GWAS study for FEV₁/FVC by UK biobank showed that the *P*-value of the *AGER* locus is not the lowest one²¹, recent larger-scaled meta-analysis of the European ancestry GWAS for FEV₁/FVC revealed that the *P*-value of the *AGER* locus is the lowest one¹⁴. Thus, our current study strongly argue that the *AGER* locus affects FEV₁/FVC similarly in the individuals of European ancestry and Japanese. We revised the descriptions.

Minor Concerns:

C3-6: Page 4 line 22 - could the authors expand on the statement that the Japanese COPD population is different to the Caucasian one?

R3-6: We agree with the reviewer about the importance to describe clinical

characteristics of Japanese COPD patients. We have added the information that the proportion of patients having emphysema-dominant phenotype is greater in Japanese COPD population than those in the US and Europe COPD population^{8,9}. We also added a sentence mentioning that the frequency of exacerbation in Japanese COPD population is smaller than those in the US and Europe COPD populations¹⁰. We have incorporated these observations into the Introduction.

C3-7: A p-value of 5×10^{-8} and 5×10^{-5} are reported. I assume these values are corrected for multiple testing and are different due to different population numbers. This needs to be clarified in text

R3-7: We appreciate the reviewer for this comment. We utilized p-value of 5×10^{-8} as the genome-wide significance level to identify the genetic locus associated with the trait (FEV₁/FVC, FEV₁ or FeNO) at Discovery Stage GWAS. To validate potential associations between the trait and the genetic locus suggested at Discovery Stage, we utilized both p-value of 5×10^{-8} and p-value of 5×10^{-5} as the genome-wide suggestive level at Validation Stage GWAS of which the size of participants is smaller than that of Discovery Stage.

This set up is based on the following assumptions. When we tried to define the threshold for the Validation Stage GWAS in terms of multiple testing for the genetic variants identified at Discovery Stage GWAS, the calculation for the threshold of each trait (0.05/ the genetic variants identified at Discovery Stage) leads us following values.

$$\text{FEV}_1/\text{FVC}: 0.05/145 \text{ variants} = 3.5 \times 10^{-4}$$

$$\text{FEV}_1: 0.05/57 \text{ variants} = 8.6 \times 10^{-4}$$

$$\text{FeNO}: 0.05/701 \text{ variants} = 7.2 \times 10^{-5}$$

Thus, the calculated threshold is relaxed compared to the genome-wide suggestive level (5×10^{-5}). Because we preferred to identify the genetic signal with a relatively stricter threshold to prevent pick up noise signals, we decided to use the genome-wide suggestive level (5×10^{-5}) for Validation Stage GWASs. We have added an explanation for these two types of thresholds in Methods.

C3-8: Table 1: Median values for population data are more informative than Mean values. Can these be changed?

R3-8: We thank the reviewer for the suggestion about Table 1. We have revised Table 1 with median values and interquartile ranges (1st quartile - 3rd quartile) instead of means and SDs.

References

1. Olin, A.C., Bake, B. & Toren, K. Fraction of exhaled nitric oxide at 50 mL/s: reference values for adult lifelong never-smokers. *Chest* **131**, 1852-6 (2007).
2. Hankinson, J.L., Odencrantz, J.R. & Fedan, K.B. Spirometric reference values from a sample of the general U.S. population. *Am. J. Respir. Crit. Care Med.* **159**, 179-87 (1999).
3. Matsunaga, K. *et al.* Reference ranges for exhaled nitric oxide fraction in healthy Japanese adult population. *Allergol Int* **59**, 363-7 (2010).
4. Zhang, Z. *et al.* Mixed linear model approach adapted for genome-wide association studies. *Nat. Genet.* **42**, 355-60 (2010).
5. Yoo, S.K. *et al.* NARD: whole-genome reference panel of 1779 Northeast Asians improves imputation accuracy of rare and low-frequency variants. *Genome Med.* **11**, 64 (2019).
6. Wyss, A.B. *et al.* Multiethnic meta-analysis identifies ancestry-specific and cross-ancestry loci for pulmonary function. *Nat Commun* **9**, 2976 (2018).
7. Kim, W.J. *et al.* Genome-wide association studies identify locus on 6p21 influencing lung function in the Korean population. *Respirology* **19**, 360-8 (2014).
8. Tatsumi, K. *et al.* Clinical phenotypes of COPD: results of a Japanese epidemiological survey. *Respirology* **9**, 331-6 (2004).

9. Izumi, T. Chronic obstructive pulmonary disease in Japan. *Curr. Opin. Pulm. Med.* **8**, 102-5 (2002).
10. Landis, S.H. *et al.* Continuing to Confront COPD International Patient Survey: methods, COPD prevalence, and disease burden in 2012-2013. *Int. J. Chron. Obstruct. Pulmon. Dis.* **9**, 597-611 (2014).
11. Park, H.Y. *et al.* Understanding racial differences of COPD patients with an ecological model: two large cohort studies in the US and Korea. *Ther. Adv. Chronic Dis.* **12**, 2040622320982455 (2021).
12. Choi, J.Y. *et al.* Clinical Characteristics of Chronic Obstructive Pulmonary Disease in Female Patients: Findings from a KOCOSS Cohort. *Int. J. Chron. Obstruct. Pulmon. Dis.* **15**, 2217-2224 (2020).
13. Lee, J.Y. *et al.* Characteristics of Patients with Chronic Obstructive Pulmonary Disease at the First Visit to a Pulmonary Medical Center in Korea: The KOrea COpd Subgroup Study Team Cohort. *J. Korean Med. Sci.* **31**, 553-60 (2016).
14. Shrine, N. *et al.* New genetic signals for lung function highlight pathways and chronic obstructive pulmonary disease associations across multiple ancestries. *Nat. Genet.* **51**, 481-493 (2019).
15. Karczewski, K.J. *et al.* The mutational constraint spectrum quantified from variation in 141,456 humans. *Nature* **581**, 434-443 (2020).

16. Hirota, T. *et al.* Genome-wide association study identifies three new susceptibility loci for adult asthma in the Japanese population. *Nat. Genet.* **43**, 893-6 (2011).
17. Steenvoorden, M.M. *et al.* The RAGE G82S polymorphism is not associated with rheumatoid arthritis independently of HLA-DRB1*0401. *Rheumatology (Oxford)* **45**, 488-90 (2006).
18. Cheng, D.T. *et al.* Systemic soluble receptor for advanced glycation endproducts is a biomarker of emphysema and associated with AGER genetic variants in patients with chronic obstructive pulmonary disease. *Am. J. Respir. Crit. Care Med.* **188**, 948-57 (2013).
19. Li, S. *et al.* Association of rs2070600 in advanced glycosylation end-product specific receptor with prognosis of heart failure. *ESC Heart Fail* **7**, 3561–72 (2020).
20. Sakurai-Yageta, M. *et al.* Japonica Array NEO with increased genome-wide coverage and abundant disease risk SNPs. *J. Biochem.*, DOI: 10.1093/jb/mvab060 (2021).
21. Wain, L.V. *et al.* Genome-wide association analyses for lung function and chronic obstructive pulmonary disease identify new loci and potential druggable targets. *Nat. Genet.* **49**, 416-425 (2017).